# S100A9-CXCL12 activation in BRCA1-mutant breast cancer promotes an immunosuppressive microenvironment associated with resistance to immunotherapy

Jianjie Li[1,2], Xiaodong Shu[1,2], Jun Xu[1,2], Sek Man Su[1,2], Un In Chan[1,2], Lihua Mo[1,2], Jianlin Liu[1,2], Xin Zhang[1,2], Ragini Adhav[1,2], Qiang Chen[1,2], Yuqing Wang[1,2], Tingting An[1,2], Xu Zhang[1,2], Xueying Lyu[1,2], Xiaoling Li[1,2], Josh Haipeng Lei[1,2], Kai Miao ⬭ [1,2,3], Heng Sun[1,2,3], Fuqiang Xing[1,2], Aiping Zhang[1,2], Chuxia Deng[1,2,3✉] & Xiaoling Xu ⬭ [1,2,3✉]

Immune checkpoint blockade (ICB) is a powerful approach for cancer therapy although good responses are only observed in a fraction of cancer patients. Breast cancers caused by deficiency of breast cancer-associated gene 1 (BRCA1) do not have an improved response to the treatment. To investigate this, here we analyze BRCA1 mutant mammary tissues and tumors derived from both BRCA1 mutant mouse models and human xenograft models to identify intrinsic determinants governing tumor progression and ICB responses. We show that BRCA1 deficiency activates S100A9-CXCL12 signaling for cancer progression and triggers the expansion and accumulation of myeloid-derived suppressor cells (MDSCs), creating a tumor-permissive microenvironment and rendering cancers insensitive to ICB. These oncogenic actions can be effectively suppressed by the combinatory treatment of inhibitors for S100A9-CXCL12 signaling with αPD-1 antibody. This study provides a selective strategy for effective immunotherapy in patients with elevated S100A9 and/or CXCL12 protein levels.

[1] Cancer Centre, Faculty of Health Sciences, University of Macau, Macau SAR, China. [2] Centre for Precision Medicine Research and Training, Faculty of Health Sciences, University of Macau, Macau SAR, China. [3] MOE Frontier Science Centre for Precision Oncology, University of Macau, Macau SAR, China. ✉email: cxdeng@umac.mo; xiaolingx@umac.mo

Breast cancer is the most common cancer in women and is one of the top causes of mortality among female malignancies worldwide. Approximately 10% of breast cancer cases are inheritable, which are associated by germline mutations of several genes, including breast cancer-associated gene 1 (BRCA1), BRCA2, p53, ATM, etc.[1,2]. BRCA1 is an important tumor suppressor gene in that: (1) germline BRCA1 mutation contributes to approximately 20%–25% of familial breast cancer cases, (2) BRCA1 expression is markedly reduced in approximately 1/3 of sporadic breast cancer cases, and (3) the majority of BRCA1-deficient cancers are triple-negative breast cancer (TNBC; ER-, PR-, and Her2-), which has the worst prognosis among breast cancer subtypes due to the lack of effective therapies[3–5]. Numerous studies have demonstrated that BRCA1 plays critical roles in many biological processes, including transcriptional regulation, centrosome duplication, regulation of multiple cell cycle checkpoints, homologous recombination-mediated DNA double-strand break (DSB) repair, DNA replication, protein stability, and mitophagy[6–16].

Because disruption of Brca1 in whole body of mouse resulted in embryonic lethality[17], we have previously generated a conditional mutant mouse model carrying with MMTV-Cre-mediated deletion of the full-length isoform of Brca1 (Brca1$^{Co/Co}$;MMTV-Cre) in the mammary tissue, and found the mutant mice developed mammary tumors after a long latency, which is accelerated in a p53 heterozygous background (Brca1$^{Co/Co}$;p53$^{+/-}$;MMTV-Cre)[8]. Because of the essential role of BRCA1 in HR mediated DSB repair, Brca1 mutant mice, and human BRCA1 mutation carriers display significantly higher tumor mutational burden (TMB) compared with BRCA1-proficient breast cancers including higher frequencies of gene mutations and extensive genomic alterations, including severe chromosomal aberrations and aneuploidy[2,5,7,9,16,18,19]. Studies have also revealed that BRCA1-deficient cancers are associated with greater numbers of tumor-infiltrating lymphocytes (TIL) than other cancers and with increased expression of immunomodulatory genes, including PDCD1 (PD-1), PD-L1, and CTLA4[20–22]. The defective HR, and/or increased expression of genes involved in immune checkpoint suggest that BRCA1-deficient cancer should have higher sensitivity to immune checkpoint blockade (ICB), as these factors are associated with increased responses to the treatment[23–26]. However, despite having these features, BRCA1 deficient breast cancers do not show a significantly improved response to ICB compared with BRCA1 proficient breast cancers[21]. These observations suggest that the factors determining the responsiveness of BRCA1-deficient cancer to ICB should be something else, yet to be identified.

To identify such determinants, here we employ CyTOF (cytometry by time of flight) followed by Data Independent Acquisition Mass Spectrometry (DIA-MS) of Brca1 deficient mammary tissues and tumors. Both approaches and further analysis demonstrate that BRCA1 negatively regulates S100A9 in mammary epithelium and the activated S100A9-CXCL12 signaling upon BRCA1 deficiency acts as a driving signaling to establish the tumor microenvironment, which renders tumor cells insensitive to ICB. This problem can be overridden by combined inhibition of S100A9-CXCL12 signaling with a PD-1 antibody (aPD-1), providing selective strategy for effective immunotherapy in breast cancers with elevated S100A9 and/or CXCL12 protein levels.

## Results

**Brca1 deficiency creates a tumor-permissive microenvironment by increasing MDSCs during tumorigenesis.** To investigate whether BRCA1 deficiency could induce defective immune response, we took advantage of our Brca1$^{Co/Co}$;MMTV-Cre mouse model (GEMM)[8] and performed immune profiling of wild-type (WT) mammary gland (WTMG), Brca1-mutant (MT) mammary gland (MTMG), tumor-adjacent mammary gland (Tu-adj. MG), wild-type tumor (WT tumor) and Brca1-MT tumors (Supplementary Fig. 1a–e). Cytobank-based tSNE analysis with CyTOF antibodies against CD45, CD3e, CD4, CD8a, CD11B, Ly6G, and Ly6C (Supplementary Fig. 1f) revealed complicated cloud images based on multiple single-cell datasets (Fig. 1a). Of note, the T cell population, including CD4$^+$ T and CD8$^+$ T cells were slightly decreased in MTMG compared with WTMG tissues, but were dramatically decreased in both Tu-adj. MG and tumor tissues (Fig. 1b). In contrast, MDSC populations, including polymorphonuclear MDSCs (PMN-MDSCs) and monocytic myeloid-derived MDSCs (M-MDSCs), were somewhat increased in MTMG tissues compared with WTMG tissues and were increased to much greater extents in Tu-adj. MG and tumor tissues in Brca1$^{Co/Co}$; MMTV-Cre mice but not in WT-Brca1 tumor-bearing mice (Fig. 1c). The increased MDSC populations in MTMG, Tu-adj. MG and tumor tissues than in WTMG and WT tumor tissues were further confirmed by immunohistochemistry (IHC) with antibodies against S100a8 and S100a9, which are the markers for MDSCs (Fig. 1d–g). These data indicate that the populations of effector cells were decreased, whereas the protector cells increased, with the progression of tumorigenesis in Brca1$^{Co/Co}$;MMTV-Cre mice.

In order to further depict the character of the development of myeloid cells between WT and Brca1$^{Co/Co}$;MMTV-Cre mice, CD11B$^+$/GR1$^+$ cells were isolated from mammary gland (MG), spleen (SP), and tumors (TM) of Brca1 WT and MT mice for RNA-sequence (Fig. 1h and Supplementary Data 1). PCA analysis showed that the character of CD11B$^+$/GR1$^+$ cells from Brca1-MT mice was quite distinct from WT mice (Fig. 1i). Further examination revealed that CD11B$^+$/GR1$^+$ cells from both spleen and mammary tissues in Brca1$^{Co/Co}$;MMTV-Cre mice had MDSC featured with MDSC signature genes revealed by Gene Set Enrichment Analysis (GSEA) (Fig. 1j, k and Supplementary Fig. 1g). Consistent with this notion, M1 like macrophages with double-positive markers of F4/80 and CD86 only showed more infiltration in Brca1-MT mammary glands (MTMG) than that in WT (p > 0.05), but significantly decrease in Brca1-MT tumor adjacent, and tumor mammary tissues compared to WT mammary gland (WTMG) (Supplementary Fig. 1h, j). In contrast, M2 like macrophages with double-positive markers of F4/80 and CD206 were increased much more in Brca1-MT mammary gland, tumor adjacent and tumor tissues than that in WT mammary gland and tumor tissues, respectively (Supplementary Fig. 1i, k). These results suggest that some intrinsic mediators secreted from Brca1-MT epithelial cells enter circulatory system and induce the accumulation of MDSCs.

One of the important functions of MDSCs is to inhibit the proliferation of T cells[27]. To test whether the increased MDSC populations in Brca1-MT tissues could inhibit T cell proliferation, we cocultured MDSCs isolated from spleen, mammary gland, and breast tumor tissues from both WT and Brca1-MT mice with T cells from 2 months-old WT mice to monitor the T cells proliferation. The data showed that MDSCs in the spleens, mammary glands, and breast tumor tissues from Brca1-MT mice had similar inhibition to the proliferation of T cells and MDSCs from both spleen and tumor tissues in Brca1-MT mice had more suppressive effects on the proliferation of T cells than those from Brca1-WT mice (Fig. 1l–o). These data suggest that Brca1 deficiency directs an immunosuppressive microenvironment characterized by markedly increased functional MDSCs with inhibition effect, which suppress effector T cell proliferation during tumor initiation and progression.

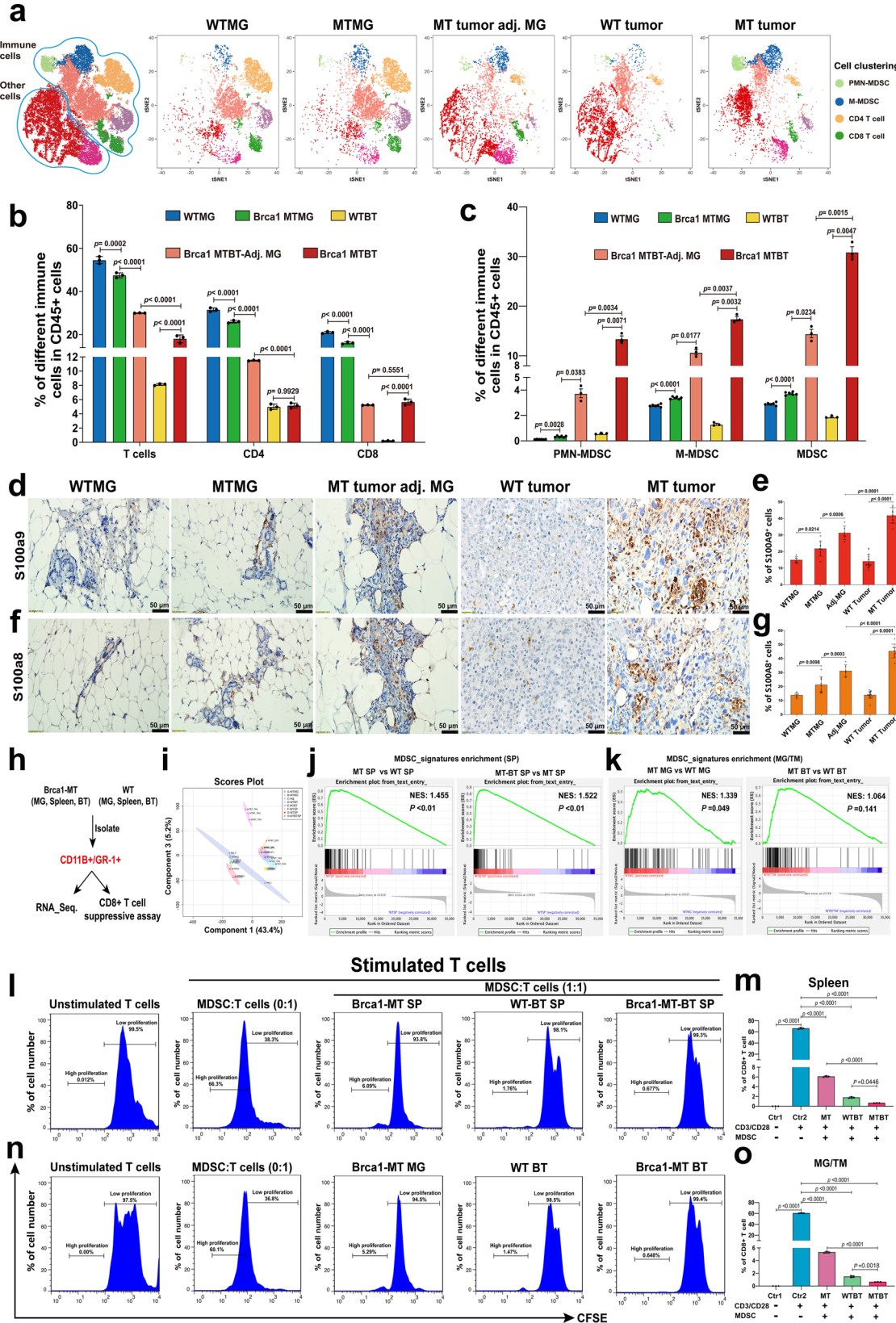

## Identification of S100A9 as an intrinsic factor in Brca1 mutant mammary tissues.

To identify intrinsic factors that contribute for enrichment of tumor permissive microenvironment, we conducted analyses with Data-independent acquisition-Mass Spectra (DIA-MS) to mammary glands of 8–10 months wild-type (WTMG), $Brca1^{Co/Co};MMTV-Cre$ (MTMG), tumor adjacent mammary gland (Tumor-adjacent MG), and breast tumor tissues from both WT (WTBT) and $Brca1^{Co/Co};MMTV-Cre$ (Brca1-MTBT) at different developmental stages of tumorigenesis in mice (Fig. 2a and Supplementary Data 2). The PCA analysis showed that different group samples were well separated (Fig. 2b). To test data reproducibility, we examined coefficient variations

**Fig. 1 Brca1 deficiency induces immunosuppression in mammary glands and tumors. a** tSNE analysis of immune cells from wildtype (WTMG, $n = 6$ mice), Brca1-mutant mammary glands (MTMG, $n = 6$ mice), Tu-Adj. mammary tissues (MT tumor adj. MG, $n = 3$ mice) from Brca1-MT mice, WT breast tumor (WT BT, $n = 3$ mice) and breast tumor (MT BT, $n = 3$ mice) from Brca1-MT mice. **b, c** The cell populations were classified as total T-cells (CD45+CD3e+), CD4+ T cells (CD45+CD3e+CD4+ and CD8-), CD8+ T cells (CD45+CD3e+CD4-CD8+), MDSCs (CD45+CD11b+Gr1+), PMN-MDSCs (CD45+CD11b+Ly6G+Ly6C-), and M-MDSCs (CD45+CD11b+Ly6G-Ly6C+). **b** Quantifications of total T-cells, CD4, and CD8 cells. **c** Quantifications of PMN-MDSCs, M-MDSCs, and total MDSCs by FlowJo analysis (Cytof gating strategies see in Supplementary Fig. 8a, $n = 3$–6 mice same in **a**). **d–g** Representative images of IHC staining with antibodies of S100a9 (**d**) or S100a8 (**f**), and quantifications of S100a9 (**e**) or S100a8 (**g**), ($n = 6$–15 pictures from 3 mice/group, 6 pictures from WTMG, 9 from MTMG, 6 from adj.MG, 15 from WTBT, and 9 from MTBT, the number of each group same in **e** and **g**). **h** The strategy of isolating CD11b+/GR1+ cells from mammary gland (MG), breast tumor (BT), and spleens (SP) for RNA-sequence and CD8 + T cells suppressive assay from the mice in **a**. **i** Principal Component Analysis (PCA) analysis of RNA-sequence data of CD11B+/GR1+cells in **h** ($n = 3$ mice/group). **j** MDSC signature genes enrichment in spleen by GSEA analysis by comparing gene expression from spleens of Brca1-MT (MT SP) vs WT (WT SP) mice and Brca1-MT mice with breast tumor (MT-BT SP) vs MT SP. **k** MDSC signature genes enrichment in mammary gland (MG) and breast tumor tissues (BT) by GSEA analysis by comparing gene expression from Brca1-MT mammary glands (MT MG) vs wild-type mammary glands (WT MG), and tumors from Brca1-MT (MT BT) and WT (WT BT) mice. **l** Representative CFSE flow-cytometry histograms and statistics from co-culture of WT T-cells with MDSCs from spleens. **m** Quantifications of % of CD8 + cells in each sample in **l**. The T-cell were from 2 months WT mice and MDSCs from spleens of 10-month Brca1-MT mice (Brca1-MT SP), Brca1-WT tumor-bearing mice (WT-BT SP) and Brca1-MT tumor-bearing mice (Brca1-MT-BT SP). Ratio of MDSC to T cells was 1:1 ($n = 3$ mice/group). **n** Representative CFSE flow-cytometry histograms and statistics from co-culture of WT T-cells with MDSCs from MG and BT. **o** Quantifications of % of CD8 + cells in each sample in **n**. The T-cell were from 2 months WT mice and MDSCs from MG of 10-month Brca1-MT mice (Brca1-MT MG), breast tumor tissues of Brca1-WT tumor-bearing mice (WT BT) and Brca1-MT tumor-bearing mice (Brca1-MT BT). Ratio of MDSC to T cells was 1:1 ($n = 3$ mice/group) (FACS gating strategies see in Supplementary Fig. 8b). The data are expressed as means ± SD (**b, c, e, g, m–o**) and $P$ values determined by one-way ANOVA followed by Tukey's multiple comparisons and permutation test. Scale bar: black color, 50 μM. Source data are provided as a Source data file.

(CV)% of duplicated samples in DIA-MS acquisition for data (Supplementary Data 3). We observed that coefficient variations of the proteins from duplicate were less than 20% and median of CV% is between 0.8 and 1.21 (Supplementary Fig. 2a). We also obtained above 0.90 Pearson correlation coefficients among the duplicated sample (Fig. 2c), indicating high reproducibility of the DIA-MS workflow in the quantification of the DIA proteome data. To identify intrinsic factors, we followed the factors which had gradually increased pattern from MTMG vs WTMG, tumor adjacent MG (Adj) vs MTMG, and MTBT vs Adj. with PatternHunter analysis (Pearson correlation coefficient $r > 0.5$, adjust $p$ value $< 0.05$) to analyze the DIA-MS data (Fig. 2d and Supplementary Fig. 2b). We identified 632 proteins from the above PatternHunter analysis and 725 proteins from tumor tissues (FC > 2, $p$ value $< 0.05$) that were differentially expressed between Brca1-MT and WT mice. (Supplementary Fig. 2b).

Next, we compared the 632 and 725 protein lists with top 45 human homolog genes which are negatively regulated by BRCA1 (Fig. 2e–g and Supplementary Fig. 2b), 4 commonly shared proteins, including *S100A9*, which plays an important role in inflammation, tumor progression and metastasis[28,29], *S100A8*, a partner of *S100A9*[30], *PGLYRP1*, and *COLGALT1*, were identified. While all the 4 top candidates might be potentially important, we first investigated *S100A8* and *S100A9*, as our CyTOF data implicating their roles in TME. Because *S100A8* knockout caused early resorption of the mouse embryo[31] and analyzing *S100a9-/-* mice revealed its critical role in regulation of neutrophil recruitment during invasive pneumococcal pneumonia[32], we hypothesized that S100A9 protein might have a critical role in mediating dynamic interactions between tumor cells and TME. Consistent with above analysis, the S100A9 peptides were identified in all five groups (Supplementary Fig. 2c) and the number of identified S100A9 peptides were coordinately increased with tumor development stages in Brca1-MT mouse mammary tissues (Supplementary Fig. 2d). In the analysis of human database, we found that *S100A9* expression level was positively correlated with human breast cancer development stages (Supplementary Fig. 2e) and Pearson correlation analysis revealed a negative correlation between *BRCA1* and *S100A9* in breast cancer patients (Supplementary Fig. 2f).

To validate above observations in vivo, we first examined S100A9 protein level in two pairs of BRCA1MT and wild-type xenograft models and we detected markedly increased S100A9+ cells in BRCA1-MT than in BRCA1-WT tumors (Fig. 2h, i). In supporting this notion, higher level expression of *S100A9* has poor survival outcomes in breast cancer patients (Fig. 2j). Next, we validated the expression and protein levels of S100a9 and of S100a8 in both mammary tissues and tumors in *Brca1Co/Co;MMTV-Cre* mice by qPCR, Western blots, and IF staining. The data revealed that both mRNA (Fig. 2k, l) and protein levels (Fig. 2m, n) were elevated in Brca1-MT MG and tumor tissues compared with controls. We also found that increased S100a9 protein was colocalized with CK18 and CD206 markers on both mammary epithelial cells and myeloid immune cells in mammary tissues of BRCA1MT xenograft models (Supplementary Fig. 2g–h), suggesting that the S100A9 might be responsible in cancer development through crosstalk between BRCA1-MT cells and surrounding stromal cells in both humans and mice.

Altogether, our CyTOF and DIA analyses identified S100A9 as the top intrinsic factor which may be responsible for the formation and dynamic changes of TME to benefit breast cancer progression.

**Expression of S100A9 is repressed by BRCA1 but positively stimulated by S100A9 itself.** Because both the mRNA and protein levels of S100a9 were increased in premalignant mammary glands and tumors of Brca1-MT mice, we investigated whether the expression of *S100a9* could be regulated by Brca1. We found that the mRNA levels of both *S100a9* and *S100a8* were 4-fold and 5-fold higher in Brca1-MT (G600) cells than in WT mammary epithelial cells (B477), respectively (Fig. 3a) and the expression of *S100a9/S100a8* gradually increased with increasing doses of shBrca1 (Fig. 3b), but not in B477 and 231 cells expressing shBrca2 (Supplementary Fig. 2a–d). Conversely, *S100a9/S100a8* mRNA expressions were repressed to 20% and 30% of its original levels in control WT (B477-Ctr) cells, respectively, when mBrca1 cDNA was introduced into B477 cells (B477-mBRCA1-cDNA) (Fig. 3c). The same results were obtained in the MDA-MB-231 (231) human basal-type breast cancer cell line (Fig. 3d),

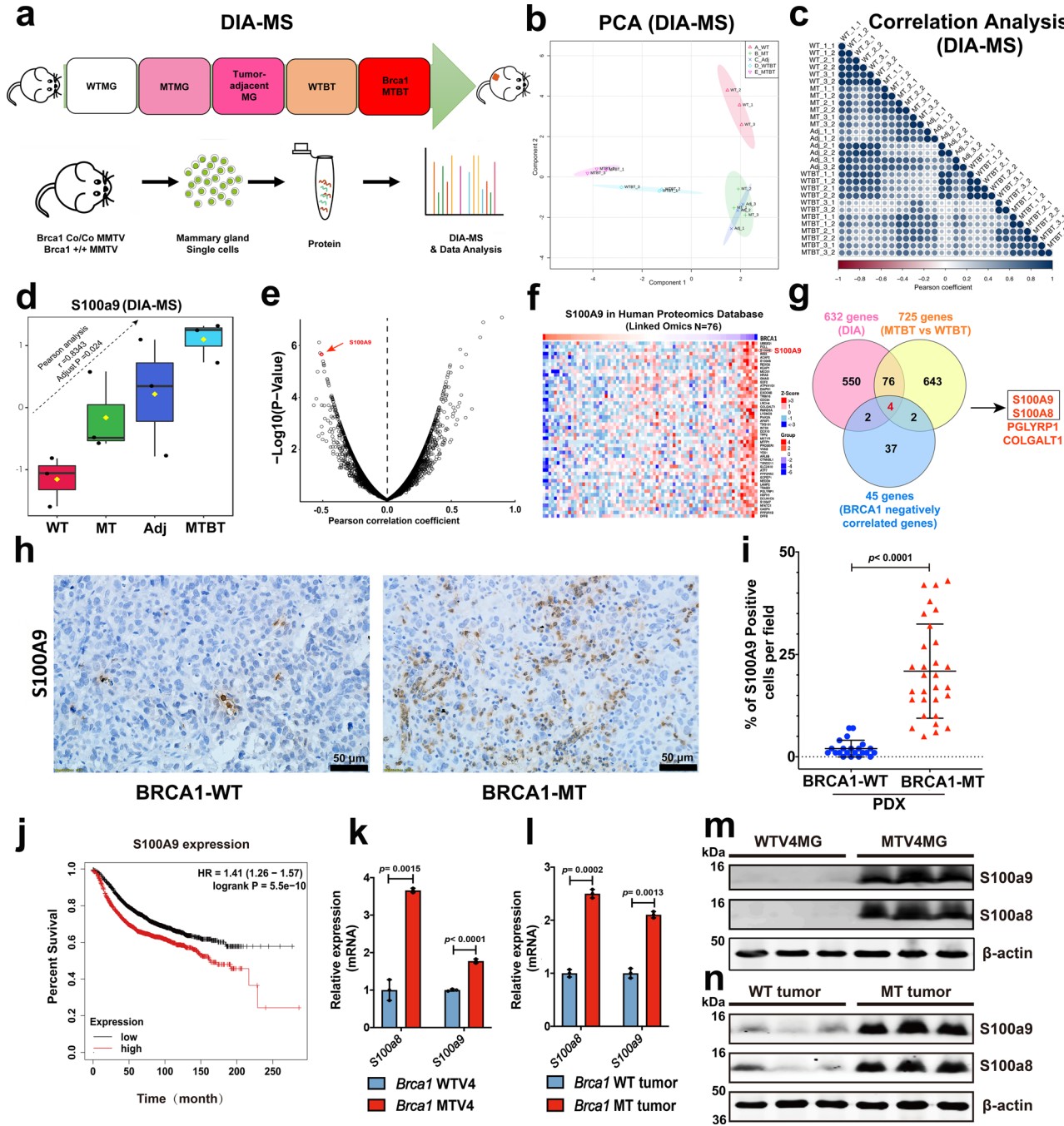

consistent with the notion that BRCA1 negatively regulates the expression of *S100A9*.

To further investigate whether *S100a9* is regulated by Brca1 at the transcriptional level, a luciferase (Luc) reporter assay (Supplementary Fig. 3e) was performed on Brca1-MT (G600) and WT (B477) mammary epithelial cell lines, which we generated earlier[11]. Promoter activity was then measured under different conditions. The data revealed that Luc activity was increased 3.8-fold in Brca1-MT cell line and to 1.5-fold in the WT cell line, respectively after 72 h of transfection with only the *S100a9* promoter-Luc (Fig. 3e). S100a9-Luc activity was down-regulated approximately to 40% in WT cells and 60% in MT cells, respectively after mBrca1 cDNA was included in the transfection (Fig. 3e), indicating a repressive effect of Brca1 on the *S100a9* promoter. Because accumulated S100a9 protein could induce expansion and activation of MDSCs and activated MDSCs can

secret S100a9[33,34], we next want to explore if *S100a9* could regulates itself to form a positive feedback loop. Of note, the Luc activity was further increased to 5-fold in Brca1-MT cells and to 3-fold in WT cells, respectively after co-transfection with S100a9-Luc promotor with m*S100a9* cDNA compared with the *S100A9*-Luc promoter transfection only (Fig. 3e). The functional domain of the promoter was determined to be in the range from −500 to −700 base pairs through serial deletion of the 2Kb *S100a9* promoter (Supplementary Fig. 3f).

To support the upregulated S100a9 signal caused by the loss of Brca1 could increase S100a9 protein levels, we knocked down BRCA1 in WT mouse mammary epithelial cells B477, and human breast cancer MDA-MB-231 cells and found that S100A9 protein levels were increased in both cell types via immunofluorescence (IF) staining (Fig. 3f) and Western blot analysis (Fig. 3g). Altogether, these data demonstrate that S100A9 expression at the

**Fig. 2 Oncogenic activation of S100A9 in both Brca1-MT mice and human breast cancer patients. a** Workflow of DIA-MS analysis with mammary tissues during tumorigenesis, including mammary gland tissues from 8–10 months Wild type (WTMG) and Brca1-MT (MTMG) virgin mice, tumor-adjacent mammary tissues (Tu-Adj. MG), and tumors from both Wild type (WTBT) and Brca1-MT (MTBT) mice ($n = 3$ mice/per genotype). **b** Plot of Principal Component Analysis (PCA) of samples from the same cohort of samples in (**a**). **c** Pearson correlation analysis of duplicate of each sample from the same cohort samples in **a**. **d** The patterns of increased the protein level of S100A9 during tumorigenesis with the same cohort of samples in **a** from DIA data analysis with Spectronaut and statistics analysis by Pearson correlation (FDR < 1, $r = 0.8343$, adjust $p$ value = 0.024. The raw data are transformed by log10,the box and whisker plots summarize the normalized values, line show the SD). The data are from $n = 3$ biologically independent replicates.
**e** Pearson correlation of BRCA1 and S100A9 at protein level by volcano plot from Clinical Proteomics Tumor Analysis Consortium (CPTAC) for The Cancer Genomic Atlas (TCGA) database (http://www.linkedomics.org/admin.php). **f** Heatmap of top 50 genes which are negatively correlated with BRCA1 at protein level from **e**. **g** Overlapping of mouse candidate gene lists from DIA analysis with top 45 genes which are negatively regulated by BRCA1 genes.
**h** The representative images of S100A9 antibody staining on BRCA1-WT and BRCA1-MT breast cancer tissues ($n = 2$ in each group from PDX models) were obtained from Jackson Laboratory. **i** Quantifications of S100A9 positive cells per field in **h**, ($n = 22$ pictures from WT and 30 from MT in h models).
**j** Survival outcome based on *S100A9* expression of breast cancer patients ($n = 4934$) from Kaplan–Meier Plotter website (https://kmplot.com/analysis/index.php?p=service&cancer=breast). **k, l** Relative expression of *S100a9* and *S100a8* of mammary tissues in 4-months Brca1-WT and MT virgin mice (**k**) and Brca1 MT and WT mammary tumors (**l**) revealed by qPCR ($n = 3$ mice/group) and the number of dots represents biological replicates).
**m, n** Protein levels of S100a9 and S100a8 in 4-month-old mammary tissues (**m**), and in Brca1 MT and WT tumors (**n**) as determined by western blots ($n = 3$ mice/group). The data are expressed as means ± SD (**d, i, k, l**) and *P* values determined by unpaired two-tailed Student's *t* test (**i, k, l**). Scale bar: black color, 50 μM. Source data are provided as a Source data file.

---

transcriptional level is negatively regulated by BRCA1 but positively regulated by S100A9 itself via activity on the S100A9 promoter; the resulting expression changes can also be detected at the protein level.

**Progressively formation of tumor-permissive microenvironment associated with elevated S100a9 long before tumor formation in Brca1-MT mammary tissues.** It has been reported that elevated S100A9 levels have pleiotropic effects on myeloid cells and could block normal differentiation of these cells[35]. Therefore, we hypothesized that elevated S100A9 levels in BRCA1-MT mammary epithelial tissue might progressively direct immunosuppressive milieu formation and promote tumor initiation and progression in mammary tissues. To test this hypothesis, we first investigated the expression of *S100a9* and its partner, *S100a8*, in different cell subpopulations in the mammary gland, different developmental stages, and in tumor tissues. We found that the mRNA levels of *S100a8/S100a9* were increased in MTMG tissues at multiple developmental stages and tumor tissues (Supplementary Fig. 4a–c), *S100a9* expression was increased dramatically in luminal and stromal subpopulations (Fig. 4a and Supplementary Fig. 4d), and Brca1-MT mammary epithelial cell line (G600) (Fig. 3a). Consistent with the increased mRNA expression (Supplementary Fig. 4a–c and Fig. 3a), elevated S100a8/S100a9 protein levels were also observed in MT mammary epithelial cell lines (G600), mammary tissues, and tumor tissues, as revealed by western blot analysis (Figs. 4b and 2m, n). Co-staining for CK18 and S100a9 in the mammary epithelium at multiple developmental stages, tumor-adjacent and tumor tissues also revealed increased S100a9 protein levels (Fig. 4c and Supplementary Fig. 4e), indicating that loss of Brca1 stimulates S100a9 expression at both the mRNA and protein levels in mammary tissues.

S100A9 is a chemoattractant[36]. Therefore, to investigate whether elevated numbers of S100a9-positive cells were present in MTMG tissue, we conducted fluorescence-activated cell sorting (FACS) analysis on both blood and mammary tissues. The data revealed that there were slightly increased S100a9-positive cells in the blood and mammary glands respectively in 4-month-old MT mice than in WT mice. Notably, the increases in S100a9-positive cell populations reached 30% in the blood and 46% in the mammary glands in 6-month-old MT mice compared to WT mice (Fig. 4d). Similar increased expression patterns were also observed for Arg1, another marker of MDSCs, in both blood and mammary gland tissues in 4/6-month-old MT mice compared to WT mice (Fig. 4d). These findings suggest that elevated S100a9

protein levels in mammary epithelial cells could drive the recruitment of additional MDSCs to MT mammary tissues, thus inhibiting the proliferation and function of cytotoxic T cells.

To further evaluate the TME in mammary tissues in Brca1-MT mice, we carried out IF staining and western blot analysis. We found that (1) S100a9-positive MDSCs invaded tumor-adjacent mammary gland tissues (Fig. 4e); (2) S100a9 protein was secreted by MDSCs into the tumor-adjacent mammary gland tissues of Brca1-MT mice and the supernatant of culture cells in vitro (Fig. 4f); and (3) the protein levels TGF-β and Il-10, which can trigger the translocation of S100A9 to the membrane and cell surface to promote S100A9 protein secretion[29,37,38], were increased in mammary tissues of both 4-month-old and 6-month-old MT mice (Fig. 4g, h). These data suggest that elevated S100A9 signaling in Brca1-MT mammary epithelial cells could serve as an intrinsic factor that causes the accumulation of MDSCs and establish a tumor-permissive microenvironment benefiting tumorigenesis in Brca1-MT mammary tissues long before tumor formation.

**Positive regulation loop between S100a9 and Cxcl12 amplifies oncogenic signals in Brca1-MT epithelial cells.** To understand the oncogenic action of *S100a9* in Brca1-MT mammary epithelial cells, we first conducted RNA sequencing on four mammary epithelial cell lines with differential expression of *S100a9*, including the WT (B477-Ctr), over expression (OE)-*S100a9*-WT (B477-OE-*S100a9*), Brca1-MT (G600-Ctr), and knock out *S100a9* (sg*S100a9*)-Brca1-MT cell lines (G600-sg*S100a9*) (Supplementary Fig. S5b–d). A total of 453 genes were upregulated in both G600-Ctr and B477-OE-*S100a9* compared with B477-Ctr cells but downregulated by G600-sg*S100a9* compared with G600-Ctr (Fig. 5a). KEGG pathway analysis of the functions of these 453 genes showed that the cytokine-cytokine receptor interaction pathway was one of the top 10 activated oncogenic pathways (Fig. 5b). Of note, *Cxcl12*, also known as stromal the cell-derived factor 1 (SDF1), was associated with these pathways and ranked first among the 453 genes in terms of its relative fold change from control level (Fig. 5c and Supplementary Fig. 5e and Supplementary Data 4). Next, we wanted to explore if Cxcl12 could mediate S100a9 signal in Brca1-MT epithelial cells. We first examined expression of 10 cytokines in B477 and G600 cells and found only *S100a9* and *Cxcl12* were up regulated in Brca1-MT cells (Supplementary Fig. 5a). This data suggested that Cxcl12 might be a main cytokine regulated by Brca1-S100a9. Then, we further investigated *Cxcl12* expression in overexpressed *S100a9* in

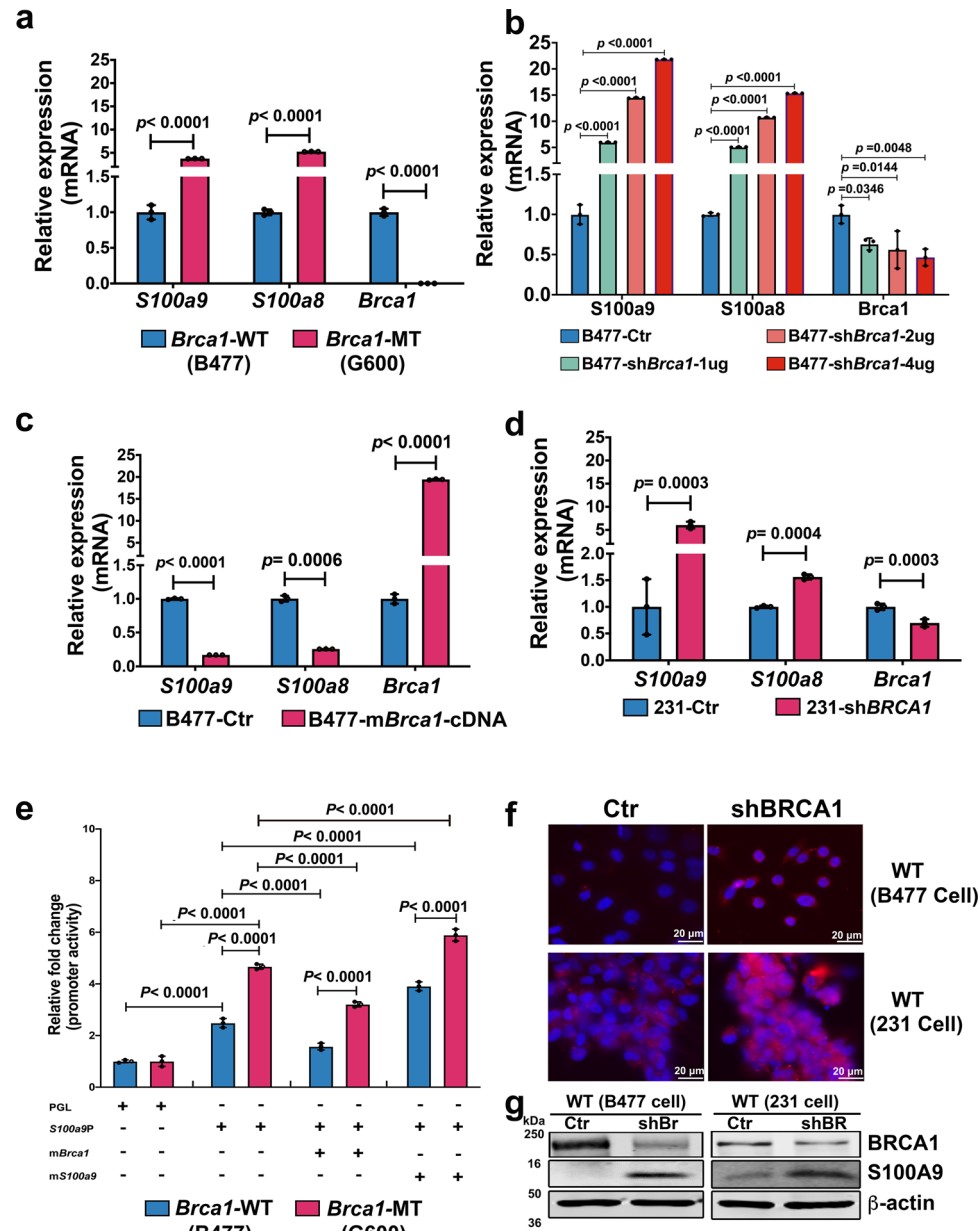

**Fig. 3 S100a9 gene is regulated by both Brca1 and S100a9. a** Expression of *S100a9* and *S100a8* in Brca1 MT (G600) and WT (B477) mammary epithelial cell lines. **b** Expression of *S100a9* and *S100a8* in B477 cells with the expression of sh*Brca1* at different concentrations. **c** Expression of *S100a9* and *S100a8* in B477 cells in which the mBrca1 gene was overexpressed. **d** Expression of *S100A9* and *S100A8* in MDA-MB-231 (231) control and 231 cells with the expression of sh*BRCA1*(knock down BRCA1). **e** Luciferase activity assay of mouse *S100a9* promoter after 72 h transfection with PGL vector only, *S100a9* promotor, *S100a9* promotor with *Brca1* cDNA, and *S100a9* promotor with *S100a9* cDNA in B477 and G600 cells. **f** S100A9 proteins in B477 mouse mammary epithelial cells and 231 cells with the expression of shBrca1 or shBRCA1, respectively by IF. **g** BRCA1 and S100A9 protein levels in B477 mouse WT mammary epithelial cells and 231 cells without or with the expression of sh*BRCA1*(shBr/shBR). The data are expressed as means ± SD (**a**–**e**) and *P* values determined by unpaired two-tailed Student's *t* test (**a**, **c**, **d**) and by one-way ANOVA followed by Tukey's multiple comparisons (**b**) or two-way ANOVA (**e**). The experiments were independently repeated three times with similar results (**a**–**g**). Scale bar: white color, 20 µM. Source data are provided as a Source data file.

B477 (B477-OE-*S100a9*) and knock out *S100a9* in G600 (G600-sg*S100a9*) cell lines (Supplementary Fig. 5b–d), and the data revealed that both the mRNA and protein levels of Cxcl12 were increased in B477-OE-*S100a9* cells but decreased in G600-sg*S100a9* (Fig. 5d, e), confirming that the expression of *S100a9* could affect *Cxcl12* expression in Brca1-MT mammary epithelial cells.

As it was shown that *Cxcl12* could induce phosphorylation of Stat3[39], therefore, we investigated whether *S100a9* could affect

pStat3 through *Cxcl12* in other cell lines. We found the protein levels of Cxcl12 and pStat3 were increased when *S100a9* was overexpressed in EMT6 cells, and such increases could be overridden by sg*S100a9* or sg*Cxcl12* in EMT6-OE-*S100a9* and B477-OE-*S100a9* (Fig. 5f–h), in 231 and 545 cells (Supplementary Fig. 5f–h). Consistent with the notion, treatment with tasquini-mod (TAS), a known S100a9 inhibitor, showed that mRNA levels of *S100a9* and *Cxcl12* were reduced in 545 and 231 cells (Supplementary Fig. 5i, j) and protein levels of S100a9, Cxcl12,

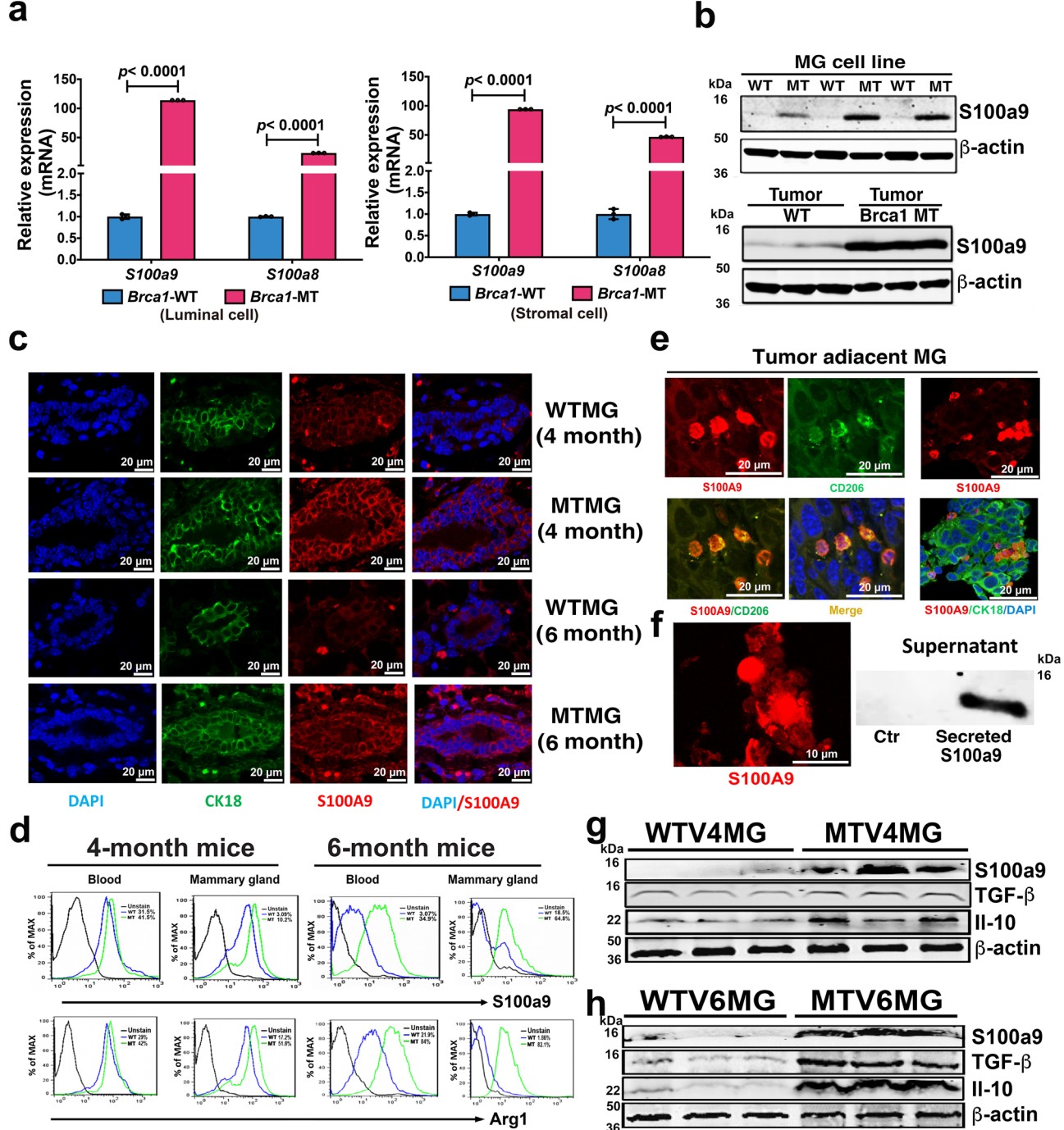

**Fig. 4 Tumor permissive microenvironment in Brca1 MT mammary glands. a** *S100a9/S100a8* mRNA expression in the subpopulations of luminal and stromal cells of WT 4-month mammary gland (WTV4MG) and MT 4-months-old virgin mammary gland (MTV4MG) (*n* = 3 mice). **b** Protein level of S100a9 in both WT (B477) and MT (G600) mammary epithelial cell lines and tumor tissues by Western blots (*n* = 3 individual experiment-up and *n* = 3 mice-down). **c** Co-staining of S100a9 (red) and CK18 (green) with antibodies on WTV4MG, MTV4MG, WTV6MG, and MTV6MG tissues (*n* = 3 pairs in each group, Scale bar: 20 μM). **d** The S100a9 and Arg1 positive cell populations by FACS analysis from the blood and mammary tissues of both WT and MT mice at 4-month and 6-month, respectively (FACS gating strategies see in Supplementary Fig. 8c, *n* = 3 mice/ group). **e** Co-staining with S100a9 (red) and CD206 (green) antibodies (left panel) and co-staining with S100a9 (red) and CK18 (green) antibodies (right panel) on tumor-adjacent tissues by IF (40X confocal microscope, Scale bar: 20 μM.) (*n* = 3 mice and 3 individual experiment). **f** Secreted S100a9 proteins (left) from both tumor cell and MDSC cells in tumor-adjacent mammary gland (*n* = 3 mice) and present in the supernatant of cultured cancer cells (right) (*n* = 3 individual experiment, Scale bar: 10 μM). **g** Protein levels of S100a9, TGF-β, and Il-10 in mammary gland tissues of both WT and Brca1 MT mice at 4-month (*n* = 3 mice). **h** Protein levels of S100a9, TGF-β, and IL-10 in mammary tissues of both WT and Brca1 MT mice at 6-month (*n* = 3 mice). The data are expressed as means ± SD (**a**) and *P* values determined by unpaired two-tailed Student's *t* test. The experiments were independently repeated three times with similar results (**a**, **b**). Source data are provided as a Source data file.

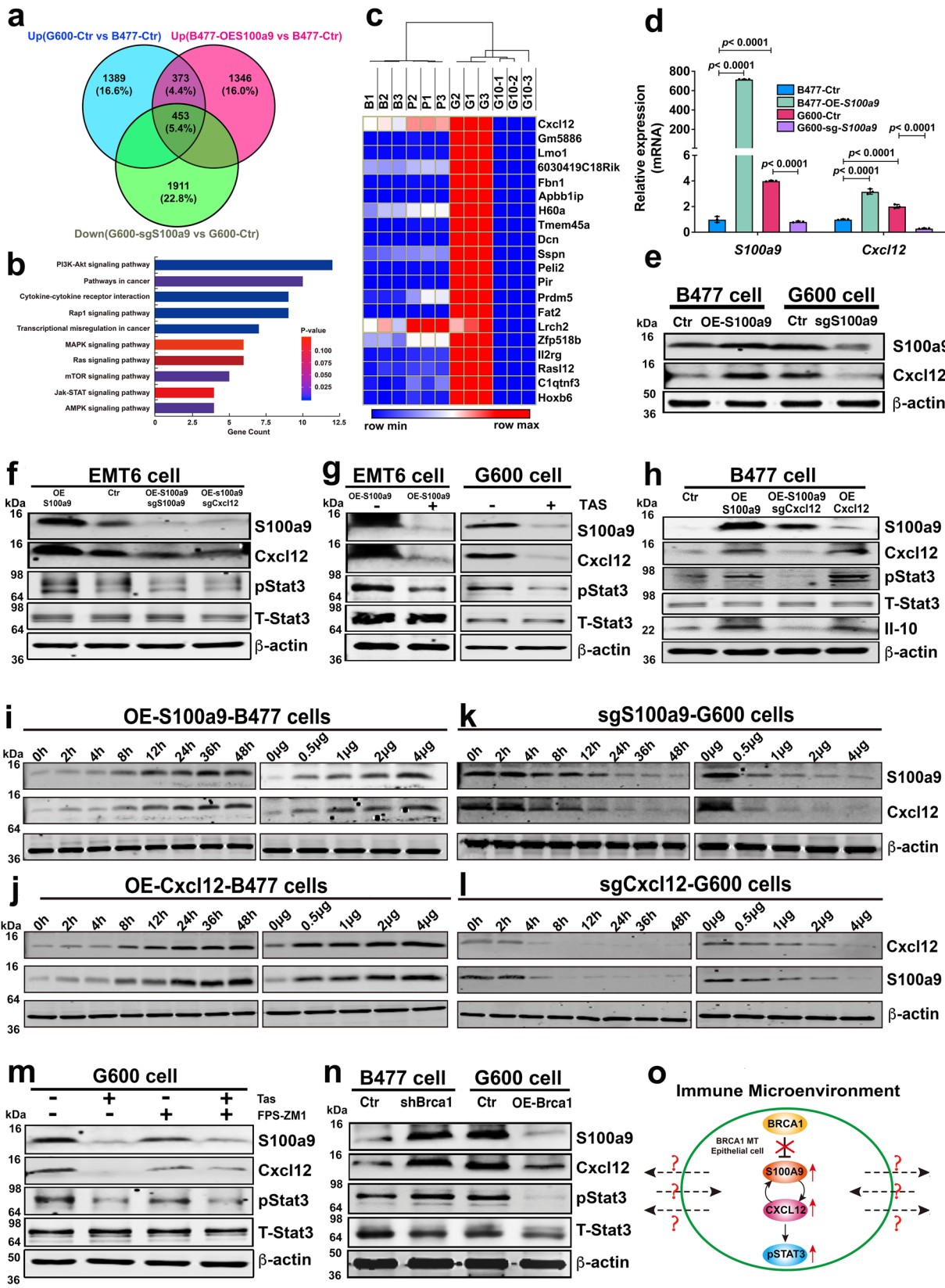

and pStat3 were decreased in G600 cells (Fig. 5g, m) which confirms a role of Cxcl12 in mediating the action of S100a9 on pStat3.

It was reported that S100A9 acts through its cognate receptor of advanced glycation end products (RAGE)[40]. To examine whether the increased pStat3 induced by S100A9 is mediated by RAGE receptor, we treated the Brca1-MT cells with FPS-ZM1, an inhibitor for RAGE[41], and detected no obvious changes in the protein levels of S100a9, Cxcl12, and pStat3. (Fig. 5m), suggesting that the regulation of Cxcl12/pStat3 by S100a9 is independent of RAGE. It has been shown that several receptors beside RAGE have been described for extracellular S100A8 and S1009,

**Fig. 5 Positive regulation loop between S100a9 and Cxcl12 amplifies oncogenic signals in Brca1-MT epithelial cells. a** The Venn diagram analysis of upregulated genes in G600, Over-Expression-*S100a9* in B477 cells (B477-OE-S100a9), and down-regulated genes in G600 cells expressing sg*S100A9* (G600-sg*S100A9*) ($n = 3$ biological replicates/group). **b** KEGG Pathway analysis with 453 common (*p* value was calculated by hypergeometric test with KOBAS 3.0 website). **c** Top 20 differentially expressed genes by comparison of the gene expression profiles of four different group cells, including B477-Ctr (B1-3), B477-OES100a9 (P1-3), G600-Ctr (G1-3), and G600-sgS100a9 (G10-1-3) (the heatmap was drawn by Morpheus website and clustered by One minus Pearson Correlation). **d**, **e** Expressions of *S100a9* and *Cxcl12* at mRNA level by qPCR (**d**) at protein levels by western blotting (**e**) from four group cells in **c**. **f** Protein levels of S100a9, Cxcl12 and pStat3 in OE-*S100a9*-EMT6 cells, sg*S100a9*/OE-*S100a9*-EMT6, and sg*Cxcl12*/OE-*S100a9*-EMT6 cells. **g** Protein levels of S100a9, Cxcl12 and pStat3 in OE-*S100a9*-EMT6 and G600 cells without or with S100A9 inhibitor, Tasquinimod (Tas-50 µM) by western blotting. **h** Protein levels of S100a9, Cxcl12, and pStat3 in WT (B477) cells with the expression of OE-*S100a9*, OE-*S100a9*/sg*Cxcl12*, or OE-*Cxcl12* by western blotting. **i**, **k** Protein levels of Cxcl12 in B477 (**i**) cells with OE-*S100a9* and in G600 (**k**) cells expressing sg*S100a9* at different time courses (0–48 h) and different amounts (0–4 µg) by western blotting. **j**, **l** Protein levels of S100a9 in B477 (**j**) cells with OE-*Cxcl12* and in G600 (**l**) cells expressing sg*Cxcl12* at different time courses (0–48 h) and different amounts (0–4 µg) by western blotting. **m** Protein levels of S100a9, Cxcl12, and pStat3 in G600 cells after treating with Tas and FPS-ZM1, inhibitor for RAGE receptors on cell membranes by western blotting. **n** Protein levels of S100a9, Cxcl12, and pStat3 in B477 cells with the expression of sh*Brca1* and MT G600 cells with over-expression of *Brca1* (OE-Brca1). **o** A diagram summarizing the relationship of Brca1, S100a9, Cxcl12 and pStat3 in Brca1-MT mammary epithelial cells, and their potential interaction with the surrounding immune microenvironment, which remains elusive. The data are expressed as means ± SD (**d**) and *P* values determined by one-way ANOVA followed by Tukey's multiple comparisons. The experiments were independently repeated three times with similar results (**d–n**). Source data are provided as a Source data file.

including TLR 4, the scavenger receptor CD36, and EMMPIRIN[42]. Whether or not these receptors play a role in S100a9-Cxcl12-pStat3 signaling is currently unclear but could be studied in future study.

Of note, our data also revealed that knockout of *Cxc12* reduced S100a9, whereas the protein level of S100A9 was moderately increased in overexpression of *Cxcl12* compared with control group in B477 (Fig. 5h and Supplementary Fig. 5f), and the effect of increased protein levels of S100A9 induced by overexpression of *Cxcl12* compared with control were observed in both B477 and 231 cells (Fig. 5j and Supplementary Fig. 5l), suggesting that there might be a positive feedback loop between S100a9 and Cxcl12. To further validation of this positive feedback loop, we performed time course and dose-dependent experiments to monitor the dynamic changes of both Cxcl12 and S100a9 in B477, G600, and MDA-MB-231 cell lines. The data reveled that over expression (OE) of either S100a9 or Cxcl12 in B477 and MDA-MB-231 cells could increase protein levels of Cxcl12 and S100a9 (Fig. 5i, j and Supplementary Fig. 5k, l). In contrast, expression of sgS100a9 and sgCxcl12 in G600 decreased protein levels of Cxcl12 and S100a9, respectively, (Fig. 5k, l), demonstrating that oncogenic signals induced by elevated S100a9 could be further amplified by the positively regulating loop between S100a9 and Cxcl12 during tumorigenesis.

To provide further evidence that the S100a9-Cxcl12 interaction acts with the loss of Brca1, we expressed knockdown Brca1 (shBrca1) in WT cell lines (shBrca1-B477) and overexpression Brca1 in Brca1-MT (OE-mBrca1-G600) cell lines. The protein levels of S100a9, Cxcl12, and pStat3 were increased in shBrca1-B477 cells and decreased in OE-mBrca1-G600 cells (Fig. 5n). Altogether, our data demonstrate that oncogenic activation of S100a9 induced by Brca1 deficiency could up-regulate Cxcl12 and activate pStat3 and up-regulated Cxcl12 protein could further amplify S100A9 oncogenic signal in Brca1-MT mammary epithelial cells, which may, in turn, interact with their surrounding immune microenvironment in a paracrine fashion (Fig. 5o). However, the underlying mechanisms and the details regarding such an interaction are currently unclear.

**S100a9/Cxcl12 axis signals from epithelial cells to immune cells**. To examine whether the S100a9/Cxcl12 signaling from the *Brca1*-MT mammary epithelium could affect the MDSC cells isolated with CD11b and Gr1 antibody, we first examined the actions of the MDSC cells from both WT and Brca1-MT mice in different conditional medium (Fig. 6a–c). The data revealed that (1) MDSCs in the conditional medium from B477 cells with

OE-S100a9 migrate more through the membrane than in their control medium from B477 cells only but more MDSCs from Brca1-MT migrate through than that from WT mice, (2) even though the same migration patterns were observed in both WT and Brca1-MT mice as above in the conditional medium from G600 cells, but much more cells migrated through the membrane from Brca1-MT mice, and (3) if the conditional medium was from G600 cells expressing sgS100a9, the number of migrating cells were significantly reduced, showing that the secreted S100A9 protein from cells with different genotype not only had effect on the migration of MDSCs but much stronger effect on MDSCs from Brca1-MT mice (Fig. 6).

To investigate how the S100a9/Cxcl12 signaling affect myeloid cells, we cultured cells of RAW264.7 cell line, a murine leukemic monocyte/macrophage cell line[43], with various types of conditioned medium (CM) for 48 h and monitored the proliferation, migration, and molecular actions of the macrophages. While CM from WT mammary epithelial cells (B477) had very little effect, CM from MT mammary epithelial cells (G600) significantly increased the colony size and migration of RAW264.7 cells. This effect was primarily caused by S100a9, as it was significantly minimized by knockout of *S100a9* in G600 cells and enhanced by overexpression of *S100a9* in B477 cells (Fig. 6d–g). Increased colony size and migration of RAW264.7 cells were also observed in cells cultured with CM from cells transfected with shBrca1 (shBrca1-WT), and reduced colony size and decreased cell migration were detected among RAW264.7 cells cultured in CM from WT cells expressing mBrca1 cDNA (OE-*Brca1*-WT) (Supplementary Fig. 6a–c). Consistent with these observations, the protein levels of genes responsible for macrophage proliferation (Cyclin D1), expansion (pStat3), and immunosuppression (Arg1)[44,45], and Cxcl12 were increased in RAW264.7 cells cultured with CM from B477-OE-*S100a9* cells and G600 cells but decreased in RAW264.7 cells cultured with CM from G600-sg*S100a9* cells (Fig. 6h). The action of the S100a9 signal from epithelial cells to macrophages was further supported with increased levels of S100a9, Cxcl12, and Arg1 proteins in RAW264.7 cells inoculated with CMs from B477 cells, which was enhanced by the presence of S100a9 protein and decreased by S100a9 inhibitor Tas (Fig. 6i). Similar data was observed using CM of EMT6 cells (Fig. 6j). These findings suggest that S100a9 could act as a signal from Brca1-MT mammary epithelial cells and WT breast cancer cells to immune cells.

To support Cxcl12 could also mediate the effects of S100a9 in immune cells, we examined how CMs from Brca1-MT mammary epithelial (G600) cells and OE-*S100a9*-EMT6 cells control

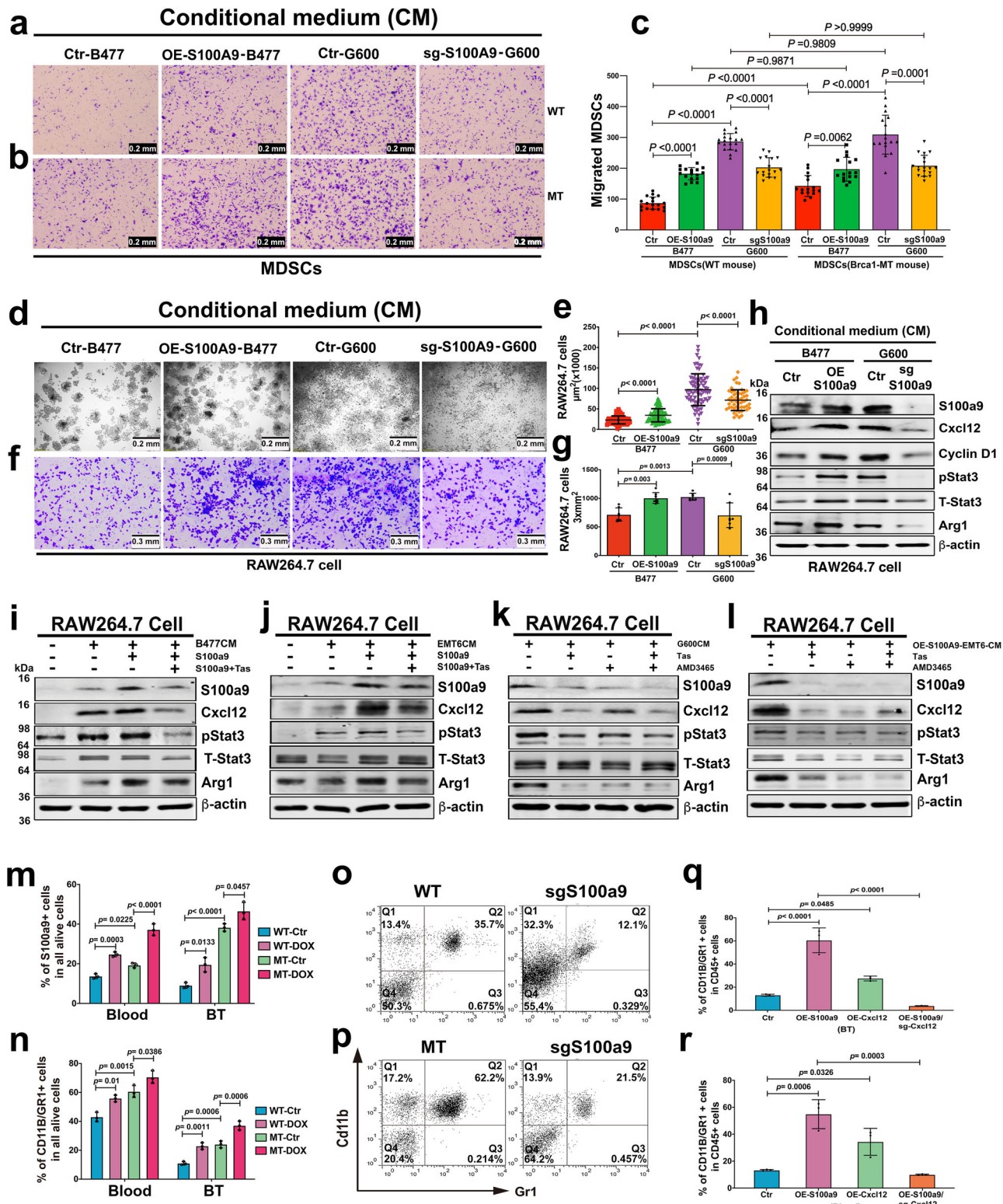

molecular actions in RAW264.7 cells. We detected reduced protein levels of S100a9, Cxcl12, pStat3, and Arg1 in RAW264.7 cells cultured with CM from Brca1-MT cells that were treated with Tas, AMD3465 a chemical inhibitor of Cxcl12 receptors or Tas together with AMD3465 (Fig. 6k). Similar effects were observed in RAW264.7 cells cultured with CM from OE-*S100a9*-EMT6 cells (Fig. 6l). These data provide compelling evidence that Cxcl12 mediates S100a9 signal from epithelial cells to immune

cells. Of note, the inhibition of Cxcl12 also diminished S100a9, showing that the same positive feedback loop of S100a9-Cxcl12 axis in immune cells.

To elucidate whether the elevated S100a9 signaling in cancers could increase MDSC accumulation in mice, we generated doxycycline (DOX) S100a9-inducible B477 cells (Brca1-WT-Dox) and G600 cells (Brca1-MT-Dox) (Supplementary Fig. 6d, e). We then implanted these cells into the mammary fat pad of nude

**Fig. 6 Expansion and accumulations of MDSCs in Brca1 MT in vivo and in vitro. a–c** Migration assay of MDSCs from spleens of WT (**a**) and Brca1-MT (**b**) mice in conditional mediums (CM), including CMs from B477 cells (Ctr-B477), OE-*S100a9* in B477 cells (OE-*S100a9*-B477), G600 cells (Ctr-G600), and G600 cells expressing *S100a9* (sg-*S100a9*-G600). Quantification **c** in (**a** and **b**, Scale bar: 0.2 mm, n = 16–19 pictures from independently repeated three times, a panel: n = 18 from Ctr-B477, 17 from OE-S100a9-B477, 19 from Ctr-G600 and 17 from sg-S100a9-G600; **b** panel: n = 16 from Ctr-B477, 16 from OE-S100a9-B477, 17 from Ctr-G600 and 18 from sg-S100a9-G600). **d–g** Colony formation (**d**) and migrating cells (**f**) of RAW 264.7 cells in different condition mediums (CMs) of Ctr-B477, OE-*S100a9*-B477, Ctr-G600, and sg-*S100a9*-G600. Quantifications (**e**, n = 52–100 cell colony area from independently repeated three times, n = 87 cells from Ctr-B477, 100 from OE-S100a9-B477, 76 from Ctr-G600 and 52 from sg-S100a9-G600) in **d** and (**g**, n = 7 pictures in each group from independently repeated three times) in **f**, Scale bar: 0.2 mm. **h** Protein levels of S100a9, Cxcl12, Cyclin D1, and Arg1 in RAW 264.7 cells with CMs of Ctr-B477, OE-*S100a9*-B477, Ctr-G600, and sg-*S100a9*-G600 by western blotting. **i**, **j** Protein levels of S100a9, Cxcl12, pStat3, and Arg1 in RAW 264.7 cells treated with CMs from either B477 (**i**) or EMT6 (**j**) cell lines with either S100a9 protein (0.1 mg/ml) or S100a9 protein together with Tas inhibitor by western blotting. **k–l** The protein levels of S100a9, Cxcl12, pStat3 and Arg1 in RAW264.7 cells treated with CMs from G600 (**k**) and OE-S100a9-EMT6 (**l**) cells without or with Tas, or AMD3465, or Tas and AMD3465 together, respectively, by western blotting. **m**, **n** The populations of *S100a9* positive (**m**) and MDSC (**n**) cells in both blood and tumors tissues after doxycycline (DOX) induction for 48 h in WT-DOX and MT-DOX mice by FACS analysis with antibodies of S100a9 and CD11b/Gr1(n = 4 mice/group) (FACS gating strategies see in Supplementary Fig. 8d). **o**, **p** MDSC populations from the blood of tumor-bearing mice with implantation of B477 (**o**) or G600 (**p**) cells without or with the expression of sgS100a9 by FACS analysis (n = 4 mice/group) (FACS gating strategies see in Supplementary Fig. 8e). **q**, **r** MDSC from breast tumor tissues (**q**) and blood samples (**r**) in Balb/c mice with fat pad implantation of Ctr-EMT6 cells, or OE-S100a9-EMT6 cells, or OE-Cxcl12 cells, or OE-S100a9/sgCxcl12 cells by CyTOF analysis (n = 6–10 mice in each group and randomly mixed them into 3 samples to do CyTOF analysis, q panel: 6 from Ctr, 10 from OE-S100a9, 8 from OE-Cxcl12, 8 from OE-S100a9/sgCxcl12, r panel: 6 from Ctr, 10 from OE-S100a9, 8 from OE-Cxcl12, 8 from OE-S100a9/sgCxcl12, gating strategies see in Supplementary Fig. 8a). The data are expressed as means ± SD (**c**, **e**, **g**, **m**, **n**, **q**, **r**) and *P* values determined by one-way ANOVA followed by Tukey's multiple comparisons. The experiments were independently repeated three times with similar results (**a–l**). Source data are provided as a Source data file.

mice and induced S100a9 expression when tumors reached 1 cm in diameter. We detected increased S100a9-positive population and CD11b/Gr1 double-positive MDSC subpopulation in the blood and breast tumor tissues 48 h after induction of S100a9 in both WT-DOX and MT-DOX mice, respectively; yet much higher levels of these populations were observed in the MT-DOX mouse model (Fig. 6m, n), We also examined the effect of sg*S100a9* in the allograft mouse tumor models, and the data indicated that CD11b/Gr1 double-positive MDSC population in the blood was greatly reduced from 36% to 12% in B477-sg*S100a9* (WT-sgS100a9) and from 62% to 22% in G600-sg*S100a9* (MT-sgS100a9) groups, respectively (Fig. 6o, p). This finding was further supported by increased TGF-β, Arg1, and Cxcl12 protein levels in the spleen of Brca1-MT mice (Supplementary Fig. 6f). Finally, we demonstrated that sgCxcl12 could block the effect of overexpression of S100a9 in the induction of MDSCs and inhibit the tumor growth (Fig. 6q, r and Supplementary Fig. 6g). Altogether, these data uncovered an important role of S100a9-Cxcl12 in inducing MDSCs accumulation to establish an immunosuppressive/tumor-permissive environment.

**Inhibition of S100a9-Cxcl12 signaling sensitizes breast cancers to immune checkpoint blockade.** Our data to this point revealed that BRCA1 mutation in mammary epithelial cells could cause oncogenic activation through the S100A9-CXCL12 axis in the mammary epithelium, promote signaling through immune cells, and induce the formation of a tumor-permissive microenvironment. To investigate the effects of inhibition of S100A9-CXCL12 axis during tumorigenesis for precision treatment, we first orthotopically implanted EMT6 cells with or without expression of sg*S100a9* or sg*Cxcl12* into BALB/c syngeneic mice. The tumor initiation and growth were significantly inhibited by the expression of sg*S100a9* or sg*Cxcl12* in EMT6 cells (Fig. 7a, b), although there were no growth differences among the three cell lines in vitro (Fig. 7c), revealing that these genes play critical roles in the tumor-permissive microenvironment. To elucidate the effects of S100a9 and Cxcl12 on immune cells in the tumor-permissive microenvironment, we performed CyTOF analysis in groups of control, sg*S100a9*, and sg*Cxcl12* and detected greater number of CD8$^+$ T cell, CD4$^+$ T cell, and active T cell populations and reduced MDSC populations in mammary tissues expressing either sg*S100a9* or sg*Cxcl12* than that of control group (Fig. 7d),

demonstrating that disruption of either *S100a9* or *Cxcl12* signaling could stimulate immunogenicity and diminish the immunosuppressive niches in mammary tissues.

To evaluate the significance of our findings for prediction of the treatment efficacy in human breast cancer patients, we analyzed a human patient database and found the correlations between the finding of our study and the human patient datasets regarding the BRCA1 deficiency-associated action of S100A9-CXCL12 axis. Data from two different databases (GSE19783-GPL6480 and TCGA) showed that increased *CXCL12* expression was correlated with low *BRCA1* expression and high *S100A9* expression (Fig. 7e, f and Supplementary Fig. 7a–g). Notably, *CXCL12* expression was elevated much more in the population with low *BRCA1* expression and high *S100A9* expression than that in the other populations (Fig. 7e). Consistent with the expression data, the survival outcomes were much worse in the population with high expression of *S100A9* and low expression of *BRCA1* (p = 0.0008748) than that in the other populations (Fig. 7f and Supplementary Fig. 7f–g). Consistent with this observation, sg*S100A9* mediated knockdown of *S100A9*-231 cells inhibited cancer initiation and growth in the xenograft mouse model (Fig. 7g, h).

Because the Brca1-MT mammary epithelial cells and immune cells exhibited high levels of S100a9 and pStat3, which could induce over expression of PD1 and/or PD-L1[46], we decided to test potential effect of ICB using an antibody to PD-1 (aPD-1), and/or inhibition of S100A9/Cxcl12 signaling. Because about 25% *Brca1$^{Co/Co}$;MMTV-Cre* mice develop mammary tumor stochastically in about 18 months[8], it is difficult to detect primary tumors at initiation stage. Therefore, we decided to use this model for studying effects of the treatment on tumor recurrence and metastasis, which frequently occur after surgical removal of primary tumors. For this experiment, we generated >100 *Brca1$^{Co/Co}$;MMTV-Cre* female mice, mating them one or two times (for activating expression of MMTV-Cre), and monitor them for the appearance of pulpable tumors, and then resected the tumors when they reached ~1.0 cm in size. We divided these mice into 4 groups with distinct treatment conditions (Fig. 7i) and monitored the growth of recurrent tumors and metastasis in the liver and lung 42 days after the initial treatment. All mice in the control group (group 1, n = 10, PBS only) had recurrent tumors in the original gland and 6/10 mice also developed additional tumors in other mammary gland, with tumor volumes ranging from 100 mm$^3$

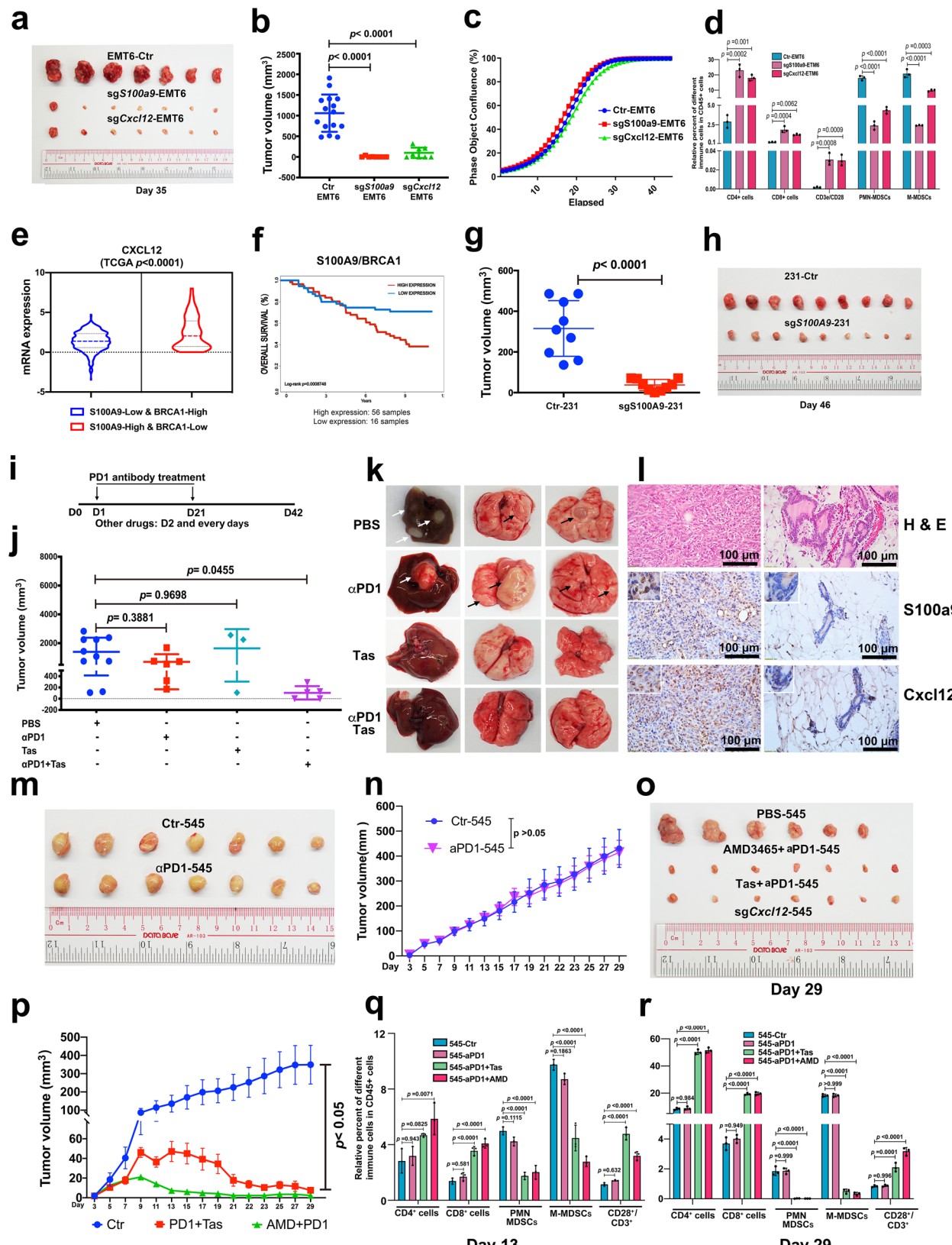

to 2250 mm³ (Fig. 7j). 8/10 mice had metastatic foci in either the liver or lung (Fig. 7k). In group 2 (aPD-1 only), 4/6 mice had recurrent tumors and also developed additional tumors in other mammary gland, with the tumor volume ranging from 150 mm³ to 1800 mm³, and all 6 mice had metastasis either in lung, or liver, or ovaries (Fig. 7j, k), indicating very little effect of ICB alone for these tumors. In group 3 (S100A9 inhibitor Tas, $n = 3$), 2/3 mice had recurrent tumors, but no metastasis was observed in either the liver or lung (Fig. 7j, k), suggesting Tas might reduce tumor metastasis although showed little effect on the growth of recurrent tumors. Of note, the data indicated that the effect of inhibitor treatment was not as good as CRISPR-Cas9 mediated knockout. We believe it is

**Fig. 7 Efficiency of combinatory treatments for breast cancer. a**, **b** Representative tumor images (**a**) and volume (**b**) from Balb/C mice after 35 days pad implantation with EMT6-Ctr cells, sgS100A9-EMT6, and sgCxcl12-EMT6 cells at 1×10⁶ cells per fat pad (n = 15 mice). **c** Proliferation of EMT6 cells and EMT6 cells expressing either sgS100a9 or sgCxcl1 in vitro. **d** The percent of different immune cells in tumor tissues without or with expression of sgS100a9 and sgCxcl12 in Balb/C mice by CyTOF analysis with antibodies to CD3/CD4, CD3/CD8, CD11b/Ly6G, CD11b/Ly6C, and CD3/CD28 (n = 3 mice/group). **e** CXCL12 expression in either BRCA1-high and S100A9-low (n = 498) or BRCA1-low and S100A9-high (n = 160) breast cancer patients from the TCGA breast database. **f** Overall survival of breast cancer patients with low expression of BRCA1 and high expression of S100A9 from the GSE19783-GPL6480 database (n = 216). **g**, **h** Tumor volumes and representative images of tumors formed from 231 cells without or with expression of sgS100A9 in nude mice (n = 9 mice/group). **i–l** Outline of treatment strategy with PBS (n = 10 mice), PD1 antibody (n = 6 mice), Tas (n = 3 mice), or PD1 antibody together with Tas (n = 5 mice) for Brca1-MT mice (**i**). Volumes and numbers of tumors (**j**), representative images of lungs and livers (**k**), and sections of the primary tumor (**l**) before treatment (left) and the residual mammary tissue of the same mouse after treatment (right). PD1 antibody was administered at day one (D1) and D21, respectively, after removing the primary tumor when the size of tumors reached 1 cm in size. Other drugs were used every day starting from D2. **m**, **n** Tumor images (**m**) and volume (**n**) at D29 from FVB mice with fat pad implantation of 545 cells and treatment with PBS and αPD1 (n = 7 mice/group). **o**, **p** Tumor volumes and representative images at the D29 from FVB mice with implantation of 545 cells (4×10⁶ cells/fat pad) and treatment with PBS (n = 6 mice), AMD3465 (AMD) + αPD1 (n = 7 mice), and Tas + αPD1 (n = 7 mice). **q**, **r** Immune cell populations in CD45⁺ cells from the same cohorts of mice in (M-P) by CyTOF analysis at D13 and D29 (n = 3 mice/group, CyTOF gating strategies see in Supplementary Fig. 8a). The data are expressed as means ± SD (**b**, **d**, **g**, **j**, **n**, **p**) and P values determined by unpaired two-tailed Student's t test (**g**, **n**) and by one-way ANOVA followed by Tukey's multiple comparisons (**b**, **d**, **j**, **p**, **q**, **r**). The experiments were independently repeated three times with similar results (**c**). Scale bar: black color, 100 μM. Source data are provided as a Source data file.

primarily because the treatment with inhibitors is affected by effective drug concentration, which usually could not reach maximum under the in vivo condition. In group 4 (Tas/αPD-1, n = 5), all mice did not have big tumor, although small residue/nodules (0–230 mm³) appeared in the original tumor site, and no metastasis was observed in either the liver or lung (Fig. 7j, k). Histology analysis of the resected breast tumors before treatment and the small nodules detected after treatment and detected reduced protein levels of S100a9 and Cxcl12 (Fig. 7l). These data suggest while ICB alone had little effect on the growth and metastasis of recurrent tumors, which is consistent with the previous finding that Brca1 deficient mouse tumors were not sensitive to ICB[21], it could be significantly enhanced by the combined inhibition of S100A9/CxcL12 signaling.

Next, we attempted to evaluate the effect of ICB/drug combinations on primary tumors taking an advantage of a tumor cell line, 545, which was derived from a Brca1Co/Co;MMTV-Cre mammary tumor and could form tumor allograft in FVB mice. The tumor growth could also be significantly minimized by CRISPR-Cas9 mediated knockout of S100a9 (Supplementary Fig. 7H). We first tested single treatment with αPD1 antibody. Consistent with the treatment to Brca1Co/Co; MMTV-Cre tumor-bearing mice, there is no inhibition to the primary breast tumor growth (Fig. 7m, n). We also found that there were increased Pdl1 and Pd1 staining in breast tumor tissues from Brca1Co/Co;MMTV-Cre treated either with Tas for S100A9 or AMD3465 which is an inhibitor for CXCR4, a receptor for CXCL12 (Supplementary Fig. 7m–p). We, next, used Tas for S100a9, AMD3465 together with αPD1 antibody to treat the FVB mice after implantation of 545 cells. The data revealed that all combinatory treatments showed to inhibit tumor growth in the first 10–20 days and nearly completely blocked tumor growth at 21 days (Fig. 7o). The treatment was stopped at day 21 and the mice were kept up to 29 days, when they were killed and tumors were analyzed for their sizes (Fig. 7p), and volumes (Supplementary Fig. 7j). Of note, all the tumor-bearing mice had enlarged spleens, which were reduced to about normal sizes after the drug treatment (Supplementary Fig. 7l).

CyTOF analysis at day13 and day29 showed that the populations of CD8⁺T cell, CD4⁺T cell, and CD3⁺/CD28⁺ active T cells were all increased but MDSC populations were decreased dramatically in combinatory treatment group compared to the control group at both day13 (Fig. 7q) and day 29 (Fig. 7r) but the effect was much more profound at day 29. Consistent with above analysis, protein levels of S100a9, Cxcl12,

and Cxcr4 in mammary tissues were dramatically reduced in αPD1-Tas and sgS100a9-545 treatment group compared to the control group and the mild reduction of Cxcl12 in AMD3465 together with αPD1 treatment group (Supplementary Fig. 7q).

All these data demonstrate that Brca1 deficiency (Fig. 8) creates a tumor-permissive environment in mammary tissues through activation of the S100A9-CXCL12 axis, which renders cancer cells insensitive to ICB, and the inhibition of S100A9-CXCL12 signaling could sensitize the cancers to ICB.

## Discussion

In this study, we investigated the impact of BRCA1 deficiency on the formation of the TME using various approaches and made several major findings: (1) BRCA1 deficiency in mammary epithelial cells increases expression of S100a9, which is secreted and positively regulates its own expression and nearby immune cells to progressively create an immunosuppressive environment in mammary tissues characterized by elevated expression of S100A9 and accumulation of MDSCs that benefits oncogenic activation in the Brca1-deficient mammary epithelium; (2) the S100A9-CXCL12 axis constitutes a positive feedback loop between the mammary epithelium and the surrounding immune environment, which enable cancer cells resistant to ICB; and (3) inhibition of S100A9 or CXCL12 sensitizes breast cancers to ICB, which may suggest a strategy for inhibiting the initiation and progression of breast cancers with elevated S100A9-CXCL12 expression.

Immune cells constitute major cell populations in TME and participate in various cancer development stages, including tumor initiation, progression, metastasis, recurrence, and drug responses[47,48]. BRCA1 is known to play important roles in many biological processes, including DNA repair, transcriptional regulation, chromatin remodeling, cell cycle checkpoint regulation, and mitophagy regulation[6–14]. However, its role in the regulation of TME formation and development remains elusive, although several recent studies have investigated this issue from various perspectives. It has been shown that BRCA1-deficient breast cancers exhibit higher abundances of tumor-infiltrating lymphocytes (TILs) and greater enrichment of T cell inflammation signatures than BRCA1-proficient breast cancers, with a wide range of variation[20,21]. However, a report also indicated that the elevated TIL abundance is only observed in BRCA1-deficient cancers that carry PTEN mutations but not with BRCA1-deficient cancers expressing WT PTEN[20]. A study on mouse has also revealed that Brca1-deficient mammary tumors with p53 mutations seem to have higher TIL levels than tumors without p53

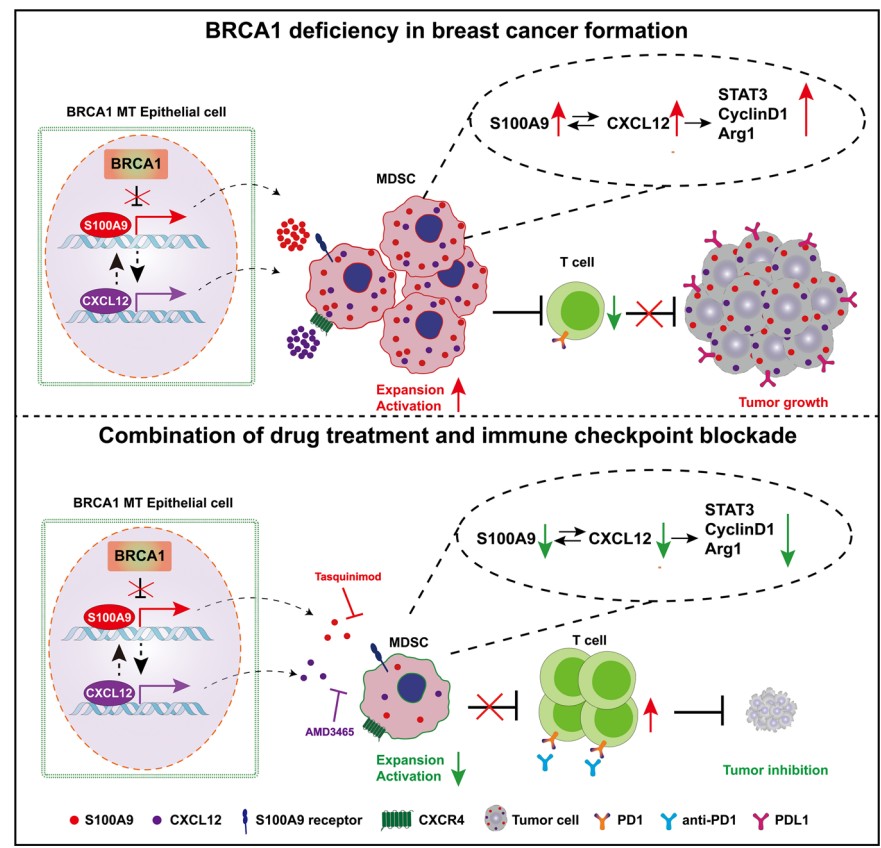

**Fig. 8 Schematic for the mechanisms by which BRCA1 deficiency in breast cancer formation and improves responses to anti–PD1 antibody.** In BRCA1 deficiency epithelial cells, S100A9 expression level constantly increases in early stages and secrets out to recruit and activate MDSCs, which creates an immunosuppression microenvironment by inhibit expansion and activation of T cells. This process can be further enhanced by CXCL12 that positively regulated by S100A9 and form a positive feedback loop. And this immunosuppression microenvironment is beneficial for tumor growth. The inhibitors of S100A9 and CXCL12, Tasquinimod and AMD3465, combine with anti-PD1 antibody can rescue the immunosuppression microenvironment and repress tumor growth. Green arrows indicate a decrease; red arrows indicate an increase.

mutation[21]. These observations may suggest that TILs are affected by factors other than BRCA1 alone. We have also recently shown that there is no obvious difference in TIL populations between Brca1-deficient mammary tumors compared with Brca1-proficient tumors[10], however abundance of immune cells in mammary tissues prior to tumor formation has not been studied. In present study, we systematically investigate this and found that CD45, CD4+ T, and CD8+ T cell populations were slightly decreased in Brca1-MT mammary gland compared with Brca1-WT mammary gland at premalignant stages. Most notably, during the course of tumor progression, MDSC populations, including both PMN-MDSC populations and M-MDSC populations, were markedly increased. MDSCs are heterogeneous immune myeloid cells that have a potent ability to inhibit the activation and function of T cells, thereby creating an immuno-suppressive environment that supports cancer growth[49]. We found that the increase in MDSC populations in BRCA1-MT mice was largely attributed to increased levels of S100A9, which is a secreted protein that can elicit effects in both intracellular and extracellular manners[50]. It has been previously shown that S100A9 is constitutively expressed in myeloid cells and plays a role in promoting myeloid cell maturation/differentiation[33,34]. Altogether, these findings suggest an intriguing working model in which loss of Brca1 in cancer cells results in upregulation of S100A9, which, in turn, not only positively regulates S100A9 expression in cancer cells but also acts extracellularly to induce MDSC accumulation. Our data also revealed that the upregulation of S100A9 was accompanied by increases in the levels of

TGFβ and IL-10, both of which are major regulators of the formation of the immunosuppressive microenvironment[51]. While the precise relationships among S100A9, TGFβ and IL-10 remain elusive, although it might be contributed by increased M2 macrophages that secrete these proteins, the increased IL-10 could also trigger translocation of S100A9 to the membrane and cell surface to promote S100A9 protein secretion[37].

However, it remains controversial in the clinical setting whether the increased expression of S100A9 predicts a good or poor prognosis of cancer patients[28,29,35,40,52]. Of note, it was shown that S100A9 activates NK cells through binding to its cognate receptor RAGE to inhibit tumor growth[40]. In our study, we found, instead of through RAGE, S100A9 positively regulates MDSC-mediated immunosuppression through CXCL12. In addition, CXCL12 also mediates the function of S100A9 in the activation of STAT3 and the immune checkpoint mediated by PD-1 and PD-L1. Although the exact mechanism for the increased levels of PD-1 and PD-L1 in BRCA1-deficient breast cancers remains elusive, we believe pSTAT3 might play an important role in this as a previous study has indicated that STAT3 activation could be responsible for the increased expression of PD-L1[46].

It has been shown previously that tumors with defective HR, and/or increased expression of genes involved in immune checkpoint are more sensitive to ICB[23–25]. However, despite having these features, BRCA1 deficient breast cancers do not show a significantly improved response to ICB compared with BRCA1 proficient breast cancers, which was shown previously[21],

and confirmed here. A most notable finding on this issue is that the resistant to ICB of BRCA1 mutant cells could be overridden by inhibition of S100a9 or CXCL12, suggesting that the activation of S100a9-CXCL12 by BRCA1 deficiency is a key determinate for the resistance. Our data indicated that inhibition of S100a9 or CXCL12 markedly increased levels of cytotoxic T cells and reduction of MDSCs.

In sum, we believe that S100A9-CXCL12 promotes tumorigenesis and tumor progression through the following actions. First, high levels of S100A9-CXCL12 promote the accumulation and amplification of MDSCs, which contributes to the induction of an immunosuppressive/anergic tumor-permissive environment to benefit tumor formation. In this regard, our data indicated that CRISPR-Cas9-mediated disruption of S100A9 or CXCL12 significantly inhibited MDSC activation in cultured cells, and in an allograft model. Consistently, treatment with a chemical inhibitor of S100A9 or CXCL12 also significantly inhibited MDSC accumulation and slowed down tumor growth. In addition, the accumulated MDSCs and increased pStat3 could activate the PD-1/PD-L1-mediated immune checkpoint to support tumor growth. Thus, we tested the efficacy of combination treatment with a chemical inhibitor of S100A9 or CXCL12 and αPD1 in BRCA1-MT mice, and found the treatment significantly inhibited mammary tumor growth, recurrence, and metastasis. This study provides a strategy to inhibit the initiation and progression of breast cancer in patients with elevated S100A9 and/or CXCL12 expression. A limitation of this study is that tasquinimod, the inhibitor for S100A9, and AMD3465, which blocks the cell surface binding of CXCL12 to CXCR4, are currently under clinical trial and has been not approved for cancer therapy yet. In addition, our study is only limited in breast cancer, future study should also be directed to determine whether inhibition of S100A9-CXCL12 signaling could also sensitize some other types of cancers to ICB.

## Methods

**Study design**. To identify intrinsic factors that are responsible for determine responsiveness of BRCA1-deficient tumors to ICB, we have employed two approaches, i.e. CyTOF (cytometry by time of flight) followed by DIA-MS, to analyze Brca1 deficient mammary tissues, and tumors derived from both BRCA1 mutant mouse models and human xenograft models. The potential candidates are validated by functional approaches including gene overexpression and CRISPR-Cas9 mediated knockout, followed by chemical inhibitors in combination of PD-1 antibody treatment for cancer therapy.

**Study approval**. All mouse strains were maintained in the animal facility of the Faculty of Health Sciences, University of Macau according to institutional guidelines. All experiments were approved by the Animal Ethics Committees of the Faculty of Health Science, University of Macau (Protocol ID: UMARE-AMEND-100).

**Tumor models and treatments**. All animal work performed in this study was approved by the University of Macau Animal Ethics Committee. Animals were maintained under a 12-h light/12-h dark cycle in a specific pathogen-free (SPF) animal facility with free access to water and a standard mouse diet. Room temperatures are maintained within the range of $23 \pm 3$ °C. The Humidity levels are controlled globally, and it is maintained between 40–70%. All animal studies were conducted on female mice, and the procedures were performed under pentobarbital sodium anesthesia. The Brca1-conditional knockout mouse model (*Brca1$^{co/co}$;MMTV-Cre*) was established in our laboratory and has been described previously[8]. To establish tumor allografts, constructed Brca1 WT (B477) mammary epithelial cell lines and Brca1-MT mammary epithelial (G600) cells ($2 \times 10^5$ cells) were implanted into the right mammary fat pads of nude mice. EMT6, sgS100A9-EMT6 and sgCXCL12-EMT6 cells ($1 \times 10^6$ cells) were implanted into the right mammary fat pads of BALB/C mice. LMBG (545) cells ($4 \times 10^6$ cells) were implanted into the right mammary fat pads of FVB mice. The tumors were measured (length and width, in mm) every other day starting on day 7. Tumor samples were collected at the end of the experiment according to the animal protocol approved by the University of Macau. The tumor volume (in mm$^3$) was calculated according to the following formula: $V = 0.5 \times H^2 \times L$, where $H$ is the shorter diameter and $L$ is the longer diameter. The tumors were measured three times per week, and samples were collected before the tumors were smaller than, or at about, 2 cm in diameter, which is the maximal tumor size/burden permitted by the University of Macau Animal Ethics Committee. In some cases, this limit has been reached the last day of measurement and the mice were immediately euthanized. For analysis of MDSCs in blood and mammary tissues, immune cells from both tissues were collected from 6-month-WT and *Brca1$^{co/co}$;MMTV-Cre* mice or from tumor-bearing nude mice when the sizes of the implanted tumors reached 0.5 cm. For drug treatment to prevent tumor growth in *Brca1$^{co/co}$;MMTV-Cre* mice, we first removed the primary tumors when the tumors reached 1 cm in size and then treated the mice with vehicle or drugs. For LMBG-FVB mice, the drugs were administered one day after injection of the LMBG cell lines. The mice were randomly assigned into a control group and drug treatment groups after removal of the primary tumors or injection of LMBG cells. The control group mice were injected intraperitoneally with PBS. For the αPD1 single-drug treatment group, αPD1 was injected twice intraperitoneally on the second day after surgery and three weeks after the first antibody injection (0.2 mg/mouse). For the AMD3465 single-drug treatment group, AMD3465 was injected every other day subcutaneously (0.2 mg/mouse). For TAS single drug treated group, TAS (an S100A9 inhibitor) was administered orally every day (0.2 mg/mouse). For combination treatment, αPD1 antibody was injected intraperitoneally every 21 days starting from the second after surgery and with AMD3465 (0.2 mg/mouse) every other day and /or TAS orally every day (0.2 mg/mouse).

**Mass Cytometry (CyTOF)**. Single cells from mouse WTMG, MTMG, adj. MG and tumor tissues were collected. Specifically, the mammary gland and tumor tissues were cut into small pieces and digested with digestion medium containing 5% FBS, 5 µg/ml insulin (I-1882; Sigma-Aldrich), 500 ng/ml hydrocortisone (H0888; Sigma-Aldrich), 10 ng/ml EGF (13247-051, Invitrogen), III 300 U/ml collagenase (S4M7602S, Worthington, Lakewood, NJ), and 100 U/ml hyaluronidase (H3506; Sigma-Aldrich) for 2 h at 37 °C. Digestion of the samples was then continued in medium containing 5 mg/ml dispase II (10295825001; Roche Diagnostics, Indianapolis, IN) and deoxyribonuclease (58C10349; Worthington, Lakewood, NJ) for 5 min at 37 °C after one wash with HPSS buffer (14170-112; Life Technologies). The tissue digests were then passed through 40 µm filters into prewarmed serum-free medium after washing with HBSS buffer. All single-cell suspensions were depleted of erythrocytes by hypotonic lysis for 2 min at room temperature (RT). The cells were resuspended to a density of $2 \times 10^7$ cells/ml in prewarmed serum-free medium, and an equal volume of 10 µM cisplatin working solution was added to the cell suspension (final concentration of cisplatin: 5 mM). Single cells were resuspended and incubated at RT for 5 min. Then, 5x the volume of the stained cell serum-containing medium was added to quench the cisplatin staining, and the cells were centrifuged at 300×g for 5 min. The cells were then washed once with 1 ml of Maxpar Cell Staining Buffer, and 3 million cells were suspended in 80 µl of Maxpar Cell Staining Buffer. The suspended cells were added to 20 µl of Fc-blocking solution. The cells were incubated for 10 min at 4 °C and resuspended in 50 µl of Maxpar Cell Staining Buffer. Fifty microliters of the antibody cocktail were added to each tube, and the cells were incubated for 30 min at RT at gentle vortex. The cells were then washed twice with 1 ml of Maxpar Cell Staining Buffer, and 1 ml of a cell intercalation solution was added (Cell-ID Intercalator-Ir diluted with Maxpar Fix and Perm Buffer). The samples were mixed well and left overnight at 4 °C. The next day, the cells were washed twice with 1 ml of Maxpar Cell Staining Buffer and then washed with 1 ml of Maxpar water. The cell concentration was adjusted to 2.5–5 × 10$^5$/ml with positive control bead buffer (EQTM Four Element Calibration (cat. #201078), and then data were acquired on a CyTOF instrument. The percentages of the different cell populations were analyzed with FlowJo 7.6.1 and GraphPad Prism 8 software. The CyTOF data were analyzed with Cytobank.

For flow cytometry detection of MDSCs, single cells were stained with antibodies against CD11b (APC, clone M1/70, BioLegend, cat. #101211), Gr1 (FITC, clone RB6-8C5, BioLegend, cat. #108419), and S100A9 (Cell Signaling, cat. #73425). Flow cytometry was performed with a standard protocol on a BD FACS Calibur$^{TM}$.

**T cell suppression assay**. T cells were isolated from the spleens of 2-month-old Brca1 WT mice using CD8a + T Cell Isolation Kit (130-104-07). MDSCs were isolated from the spleen of 10-month-old Brca1-MT mice following Myeloid-Derived Suppressor Cell Isolation Kit (130-094-538) protocols. The isolated T cells were stained with CFSE (10 µM) at RT for 15 min. Next, 4–5 volumes of cold complete medium were added, and the mixtures were incubated on ice for 5 min to quench the staining reaction. MDSCs were mixed T cell at 1:0 or 1:1 ratio and seeded onto CD3/CD28-coated 96-well plates. Then, the plates were incubated at 37 °C for 72 h, and the cells were subjected to FACS analysis.

**DIA-MS**. Fifteen samples were collected from WT mammary gland (WTMG) -virgin 8–10 month, MT mammary gland (MTMG) -virgin 8–10 month, WT breast tumors (WTBT), Brca1-MT tumors (Brca1-MTBT), and adjacent (BRCA1-MTBT adj.-MG) ($n = 3$ mice/group). The tissues were digested into single cells following above Cytof digestion single cells procedures. In this study, DIA was adopted to do mass spectrum analysis and performed by BiotechPack

SCIENTIFIC. For DIA analysis, equal amounts of samples (~5 μg protein) were injected in two technical replicates. The details of DIA-MS are as follows:

**Proteomics sample preparation**. The single cell samples from above groups (15 samples) were lysed by adding 200 μL RIPA buffer (Thermo Scientific™, #89900) and followed by 40 min sonication at 4 °C. The lysate was centrifuged at 10,000×*g* for 10 min at 4 °C, and the supernatant was collected to measure the concentration by BCA assay (Thermo Scientific™ Pierce™ BCA Protein Assay Kit, #23227). Aliquot of 50 μg of protein was digested by Trypsin. Briefly, the samples were diluted to 100 μl with 50 mM NH4HCO3 (≥99.5%, Sigma-Aldrich). Then, proteins were reduced by DTT (final concentration 10 mmol/L, Thermo Scientific™ #R0862) at 56 °C for 1 h and alkylated by Iodoacetamide (IAA-final concentration 50 mmol/L, Thermo Scientific™) in the dark for 40 min. at room temperature. Samples were diluted with 600 μL pre-cooled (−20°C) acetone and freeze at −20 °C overnight. The sample was centrifuged at 8000×*g* for 10 min. at 4 °C. Carefully poured out acetone to keep white precipitate. Let the precipitate dry for 2–3 min. The protein precipitation was redissolved with 100 μL 50 mM NH4HCO3 and digested overnight at 37 °C by 1 μg trypsin (Promega, Madison).

**Data dependent acquisition**

*Peptide separation*. Part of above digested peptides were mixed into 1 pool and were dissolved in sample solution (0.1% formic acid (FA)-Sigma-Aldrich, 2% acetonitrile (ACN)-Fisher Chemical) to 1 μg/μL, and the sample volume was 100 μL. The peptides were separated by Nano LC (Easy-nLC 1000-Thermo Fisher Scientific-USA). Specifically, 100 μg peptide sample was loaded onto the column (150 μm i.d. × 150 mm, packed with Acclaim PepMap RPLC C18, 1.9 μm, 100 Å). Mobile phase flow rate was 0.5 mL/min with A phase (2% ACN) and B phase (98% ACN). The gradient was performed as follows: 5–8% B in 0–5 min, 8–18% B in 5–20 min, 18–40% B in 20–70 min, 40–95% B in 70–72 min. Fraction collection: from the first minute after elution to the end of 72 min, the fractions collected at different times were combined into 12 components and dried under vacuum at 45 °C.

*Data dependent acquisition*. The above 12 fractions were dissolved in sample solution (0.1% FA, 2% ACN) and loaded onto Nano LC connected to an electrospray ionization mass spectrometer/ESI Mass Spectrometer (Orbitrap Eclipse Mass Spectrometer-Thermo Fisher Scientific-USA) to adopt in a data dependent acquisition (DDA) model. The specific steps and parameters were as follows: Mobile phase flow rate was 600 nL/min with A phase (0.1% FA) and B phase (0.1% FA, 80% ACN). The samples were loaded onto column (150 μm i. d. × 150 mm, packed with Acclaim PepMap RPLC C18, 1.9 μm, 100 Å.) and each fraction was analyzed 120 min. The isolated peptides were directly entered into the mass spectrometer for online detection. The MS1 scan was acquired from a range of 350–1500 *m/z* at a resolution of 70,000, followed by automatic gain control (AGC) set at 4.0e5 and maximum injection time of 50 ms. For MS2 scan, the acquired range is 200–2000 *m/z* at a resolution of 17,500, followed by AGC set at 3.0e5 and maximum injection time of 72 ms. The normalized collision energy (NCE) is 27% and Activation Type-HCD (Collision Energy 35%).

**Data independent acquisition**. The peptides of 15 samples (5 μg each) were dissolved in the sample solution (0.1%FA, 2% ACN) and adopted in a DIA model. Each sample was injected in duplicate technical replicate. The specific parameters were as follows:

The MS1 scan was acquired from a range of 400–1210 *m/z* at a resolution of 120,000, followed by a AGC set at 4.0e5 and maximum injection time of 50 ms. For MS2 scan, the acquired range is 200-2000 m/z at a resolution of 17,500, followed by AGC set at 3.0e5 and maximum injection time of 72 ms. The NCE is 27% and Activation Type-HCD (Collision Energy 35%).

**Spectral library generation**. The above 12 DDA MS produced files were imported into Proteome Discoverer (Version 2.4 Thermo Scientific) to search and identification. Search parameters were as follows: Fixed modifications: Carbamidomethyl(C); Variable modifications: Oxidation (M); Enzyme: Trypsin; Maximum Missed Cleavages: 2; Peptide Mass Tolerance: 20 ppm; Fragment Mass Tolerance: 0.6 Da; Mass values: Monoisotopic; Significance threshold: 0.05.

**DIA analysis**. Spectronaut software was used to analyze the raw data of DIA and the detailed parameters were as follows: peptide length: 7-30; Enzymes/Cleavages Rules: Trypsin/P; Digest Type: Specific; Missed cleavages: 2; Fragment Ions *m/z*: 400–1210; Max Variable Modifications: 5; Fixed Modifications: Carbamidomethyl(C); Variable Modifications: Acetyl (Protein N-term), Oxidation (M) PSM-FDR: 0.01; Peptide FDR: 0.01; Protein Group FDR:0.01; Precursor Qvalue Cutoff: 0.01; Protein Qvalue Cutoff (Run): 0.05.

**Cell lines, culture conditions and viral infection**. The WT (B477) mammary epithelial cell line was derived from the mammary gland of a Brca1-WT mouse, and the Brca1-MT (G600) mammary epithelial cell line was derived from a Brca1-MT mouse as previously described[11]. MDA-MB-231, HEK293T, EMT6,

RAW264.7 cell lines from ATCC. All cell lines in this study were maintained at 37 °C under 5% CO2 in Dulbecco's Modified Eagle's Medium (DMEM) (Life Technologies, Carlsbad, CA) supplemented with 10% fetal bovine serum (FBS) (Sigma, St. Louis, MO), 10 μg/ml insulin (Invitrogen, Carlsbad, CA), 1% L-glutamine (Life Technologies), and 0.6% Pen-Strep (Life Technologies). EMT6 cells were cultured with RPMI 1640 (Life Technologies, Carlsbad, CA) supplemented with 10% FBS, 10 μg/ml insulin, 1% L-glutamine, and 0.6% Pen-Strep. For lentiviral transduction, 293T cells were transfected with psPAX2 (3 μg) or pMD2.G (9 μg) and Transfer Plasmid (12 μg). The packaged virus was added for 72 h, and the transduced cells were selected with puromycin. BRCA1 cDNA and shRNA plasmids were transfected into cells with Lipofectamine 3000 Transfection Reagent. The sgRNA sequences used in this paper were listed in Supplementary Table 1.

**Reagents and antibodies**. Anti-CD11B (M1/70, 101211, 1:100 for million cells for FC), anti-CD3 (17A2, 100203, 1:100 for million cells for FC), anti-Gr1 (RB6-8C5, 108419, 1:100 for million cells for FC), anti-CD28 (37.51, 102109, 1:100 for million cells for FC), anti-CD4 (GK1.5, 100405, 1:100 for million cells for FC) antibodies were from Biolegend. Anti-CD8a (53-6.7, 2082909, 1:100 for million cells for FC) was from Invitrogen. Anti-Arg1 (IC5868A, 1:100 for million cells for FC) was from R&D. Anti-S100a9 (PE, D3U8M, 1:100 for million cells for FC), anti-S100a9 (#73425, 1:300 for WB), anti- p-Stat3 (4113 S, 1:1000 for WB), anti-Cyclin-D1 (2978 S, 1:1000 for WB), F4/80 (70076 S, 1:50 for IF), PD1 (84651 S, 1:50 for IF) and PDL1 (13684 S, 1:50 for IF) antibodies were from Cell Signaling. Anti-CD3 (ab5690,1:100 for IF), Anti-Arg1 (ab60176, 1:1000 for WB), anti- CXCR7 (ab117836, 1:1000 for WB), anti-IL-10 (ab189392, 1:1000 for WB), anti- CXCR4 (ab124824, 1:100 for IHC) antibodies were from Abcam. Anti-BRCA1 (SC-642, 1:200 for WB), anti-c-Myc (SC-764, 1:200 for WB), anti-TGF-β1(SC-130348, 1:300 for WB), Bcl-XL (SC-634, 1:500 for WB), and anti-CK18 (sc-53256, 1:50 for IF) were from Santa Cruz. Anti-S100A8 (157921-1-AP, 1:500 for WB, 1:50 for IHC) was from Proteintech. Anti-CXCL12 (MA5-23759, 1:300 for WB, 1:50 for IHC) was from ThermoFisher. Anti-CD206 (BD AF2535, 1:100 for IF) was from BDPharmingn. Anti-CD86 (LS-C392134, 1:50 for IF) was from LSBio.

Doxycycline (D9891), AMD3465 (85991-07-5) were from Sigma. Carboxyfluorescein succinimidyl ester (CFSE) (65-0850-84) was from eBioscienceTM. Cell-ID Cisplatin (201064), Cell-ID Intercalator-Ir (201192 A) were from Fluidigm. Puromycin (ant-pr-1) was from InvitroGen. Tasquinimod (S7617) was from Selleckchem. FPS-ZM1 (S8185) was from Selleckchem. S100A9-Recombinant protein (LS-G3612-50) was from LSBio. Dual-Luciferase® Reporter Assay System (E1960) was from Promega. Bio-Rad Protein Assay Dye Reagent Concentrate (#500-0006) was from BIO-RAD. PrimeScriptTM RT Reagent Kit (#RR037A) was from TaKaRa. Corning® BioCoatTM Matrigel® Invasion Chamber (#354480) was from Corning.

Mouse Brca1 shRNA1 (Cat#44594), shRNA2 (Cat#44595), Human BRCA1 shRNA1 (Cat#90549) and shRNA2 (Cat#90548) were purchased from Addgene. Mouse Brca1 cDNA (Cat# MG212086) was purchased from Origene. Human S100A9 cDNA (Cat#20200424), human CXCL12 cDNA (Cat#20200408), and mouse Cxcl12 cDNA (Cat#20200407) were purchased from Generay Biotech. Mouse S100a9 cDNA (Cat#MG50284-CF) was purchased from Sino Biological Inc.

**RNA isolation and real-time qPCR analysis**. RNA was isolated from fresh mammary gland and tumor tissue following as previously described protocol[11], and reverse transcription was performed using a PrimeScriptTM RT Reagent Kit with gDNA Eraser (TaKaRa #RR047Q). Real-time qPCR was conducted using SYBR-Green ER Master Mix (Roche, 24759100) and a QuantStudio 7 Flex Real-Time PCR System (Thermo Fisher). The primers used in Real-time qPCR were listed in Supplementary Table 1.

**RNA sequence**. Total RNA was extracted with TRIzol reagent (Invitrogen) and the concentration and integrity of RNA was measured by Agilent 2100 Bioanalyzer (Agilent RNA 6000 Kit). RNA was purified by the NEBNext Poly(A) mRNA Magnetic Isolation Module (NEB, E7490). Purified mRNAs were reverse-transcribed into cDNA and performed library construction using the NEBNext Ultra™ II RNA Library Prep Kit for Illumina (NEB, E7770) following the manufacturer's instructions. RNA libraries were sequenced on Illumina HiSeq 3000 (SY-401-3001). The raw sequencing data were aligned to mouse genome reference mm10 by using hisat2 (v2.1.0) with default parameters. FeatureCounts (v1.6.3) was applied to count the number of reads that mapping to each genes. Metaboanalyst online tool-MetaboAnalyst 5.0 (https://www.metaboanalyst.ca/) was used for PCA analysis. GSEA_4.1.0 was used to draw GSEA MDSC_signatures enrichment.

**Protein extraction and immunoblotting**. Tissues and cells were lysed with RIPA buffer (50 mM Tris-HCl, pH 7.4, 150 mM NaCl, 1% NP-40, 0.1% SDS, 0.5% sodium deoxycholate, 1 mM EDTA, and 10% glycerol) supplemented with a protease inhibitor and phosphatase inhibitor cocktail (Thermo Fisher Scientific). Twenty micrograms of protein were loaded and subjected to SDS-PAGE. The blots were incubated with primary antibodies at 4 °C overnight and with secondary antibodies at RT for 1 h. The band intensities were determined with an ODYSSEY CLx system. The detail was followed as previously described[11].

**IHC and IF staining**. Slides containing paraffin sections were placed in a slide holder. For deparaffinization and rehydration of the sections, the slides were placed into xylene two times for 5 s, into 100% ethanol for 1 min, into 95% ethanol for 1 min, into 85% ethanol for 1 min, into 70% ethanol for 1 min, into 50% ethanol for 1 min, into 30% ethanol for 1 min and were then rinsed with tap water for 1 min. The slides were washed with PBS three times for 5 min each time and then baked in an oven overnight in R-Buffer-A (10 ml in 90 ml of water). The slides were again washed with PBS three times for 5 min each time before being incubated in 0.5% Triton X-100 in PBS at 37 °C for 5 min, washed with PBS three times for 5 min each time, treated with 0.5 mg/ml of sodium borohydride (in PBS) at RT for 10 min, washed with PBS three times for 5 min each time, incubated with blocking solution (50% 3% BSA and 50% Animal-Free Blocker) overnight, washed with PBS three times for 5 min each time, incubated with primary antibodies at 4 °C overnight, washed with PBS three times for 5 min each time, incubated with secondary antibodies at RT while protected from light, and washed once more with PBS three times for 5 min each time. A drop of water-based mountant and a coverslip were placed on each slide. The slides were stored in a slide box to prevent fading, and images were captured with a microscope.

**Luc assay**. Cells (0.2 million) were seeded onto a 24-well plate, and 0.8 μg of plasmid was added to each well the next day following the protocol. After 72 h, the growth medium was removed from the cultured cells, and the cells were rinsed in 1x PBS. A Luc assay was performed with a kit (Promega-E1500) following the manufacturer's protocol. Then, 100 μl of cell culture lysis buffer (1x PLB) was added per well, and the cells were shaken at RT for 15 min. Twenty microliters of cell lysate was added to the 96-well plate for Luc measurement. The Luc readings were obtained after addition of 100 μl of Luc Assay Reagent II, and the Ren readings were obtained after addition of 100 μl of G10 Stop Reagent. The results are expressed as the Luc/Ren ratios.

**FACS**. Single cells were collected and washed once with PBS. Approximately 5% of the total cells were used as negative controls, and the remaining cells were used for the experiment. The remaining cells were resuspended in 100 μl of PBS, mixed gently, and incubated with 1 μl of antibody on ice for 30 min in darkness. The cells were washed once with 1 ml of PBS and centrifuged for 5 min at 300×g at 4 °C. The supernatant was discarded, and the cell pellet was resuspended in 100 μl of PBS. The resuspended cells were then incubated with a secondary antibody (1:100) on ice for 30 min, washed once, and brought to a final volume of 200 μl with PBS. The solution was passed through a 40 μm filter, and FACS was performed.

**IF staining of cultured cells**. Cells were seeded according to the protocol, washed with PBS two times and fixed with 4% formaldehyde for 30 min. The cells were then washed with PBS three times for 5 min each time, treated with 0.25% Triton X-100 for 20 min, washed with PBS three times for 5 min each time, blocked with blocking solution for at least 30 min, incubated with primary antibodies overnight at 4 °C, washed with PBS three times for 5 min each time, incubated with secondary antibodies with DAPI for 1 h at RT, and washed with PBS three times for 5 min each time. The slides were air-dried, covered with coverslips, and imaged under a microscope.

**Cell proliferation assay**. A total of 5000 cells per well were seeded onto 96-well plates according to the experimental design, and the plates were placed into an IncuCyte® Live Cell Analysis System for culture for 1 week. Proliferation curves were then drawn.

**Transwell assay and conditional medium assay**. The kit was removed from −20 °C storage and allowed to come to RT. Next, 0.5 ml of warm (37 °C) DMEM was added to the interiors of the inserts and the bottoms of the wells. The materials were allowed to rehydrate for 2 h in a cell culture incubator at 37 °C under 5% CO₂. After rehydration, the medium was carefully removed without disturbing the layer of Corning Matrigel Matrix on the membrane, and the designated CM was added to the wells of the Falcon TC Companion Plate. Next, the chambers were transferred to the wells containing the CM, and 0.5 ml of cell suspension ($5 \times 10^4$ cells/ml of culture medium) was added to each well of the 24-well chambers. The Corning BioCoat Matrigel Invasion Chambers were incubated for 22 h in a cell culture incubator at 37 °C under 5% CO₂. After incubation, the non-invading cells were removed from the upper surface of the membrane. The cells were stained with 0.05% crystal violet (1% methanol, 1% Formaldehyde) for 20 min in RT and imaged.

**Statistical analysis**. For quantification of tumor cell or immune cell density, images of tumor sections with IF staining were captured and analyzed with ImageJ software. Statistical analysis was performed by using GraphPad Prism 8.0. Unpaired two-tail Student's $t$ test were used for comparison of differences between two groups. For comparisons between multiple groups, one-way ANOVA or two-way ANOVA were used. Correlations were analyzed with two-tailed Pearson tests. Statistical information, including the means and statistical significance values, is indicated in the text or figure legends. On graphs, bars represent either range or standard deviations (SDs), as indicated in legends. The survival benefits for the animal and human datasets were determined by Kaplan-Meier analysis. The DIA-MS data were analyzed by PatternHunter analysis of the website (https://www.metaboanalyst.ca/faces/home.xhtml) and genes were filtered by Pearson $r > 0.5$, adjust $p$ value< 0.05. The filter thresholds of upregulated and downregulated genes for RNA sequencing data were a fold change >1.2 or <0.83 respectively, and $p$ value < 0.05. The websites used in this paper were listed in Supplementary Table 2.

**Reporting summary**. Further information on research design is available in the Nature Research Reporting Summary linked to this article.

## Data availability

The RNA-seq data have been deposited at Sequence Read Archive (SRA) database with following access numbers: PRJNA766531, PRJNA719246 and PRJNA719077. The DIA-MS proteomics data have been deposited at PRoteomics IDEntifications Database (PRIDE) with following access numbers: PXD030328 (DIA raw data) and PXD030355 (DDA raw data). Breast cancer data from The Cancer Genomic Atlas [https://portal.gdc.cancer.gov/projects/TCGA-BRCA] (TCGA-BRCA.sampleMap/HiSeqV2) database and GSE19783-GPL6480 dataset was used. The remaining data are available within the Article, Supplementary Information or Source Data file. Source data are provided with this paper.

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

## Acknowledgements

The authors thank the members of the Xu laboratory for helpful advice and discussion; the Animal Research Core for providing the animal housing and Genomics, Bioinformatics and Single Cell Analysis Core for providing CyTOF analysis service. This work was supported by a multi-year research grant (MYRG2016-00138-FHS, MYRG2017-0008-FHS, MYRG2019-0064-FHS) by the University of Macau, Macau SAR, China; The Science and Technology Development Fund (FDCT) grants (027/2015/A1, 029/2017/A1, 0101/2018/A3, and 0011/2019/AKP, 0097/2021/A2) of Macau SAR, China.

## Author contributions

Conception and design: J.L., C.D., and X.X. Development of methodology: J.L., X.S., and J.X., Acquisition of data (provided animals, acquired and managed patients, provided facilities, etc.): J.L., X.S., J.X., S.M.S., U.I.C., K.M., Q.C., X.Z., L.M., J.L., Y.W., T.A, H.P.L, H.S, F.X., and X.L. Analysis and interpretation of data (e.g., statistical analysis, biostatistics, computational analysis): J.L., J.X., R.A., X.Z., X.L., A.Z, C.D., and X.X. Writing, review, and/or revision of the manuscript: J.L., X.X., and C.D.

## Competing interests

The authors declare no competing interests.
