## [Peer Review File · Nature Communications]

Reviewers' Comments:

Reviewer #1:

Remarks to the Author:

The manuscript entitled "S100A9-CXCL12 axis in BRCA1 deficiency orchestrates the immunosuppressive microenvironment to enable breast cancer resistant to immune checkpoint blockade" identifies a novel connection between BRCA1 loss in breast cancer and elevation of S100A9 and CXCL12 providing a possible mechanism for checkpoint resistance. Upon identifying S100A9, the authors confirm the increased expression in their BRCA MT system and connect elevation of S100A9 to patient data. The authors then examine the BRCA/S100A9/CXCL12 axis in a variety of cell lines with using both CRISPR and OE of the proteins. Furthermore, the authors utilize a variety of in vivo models to provide additional evidence of the BRCA/S100A9/CXCL12 axis. Overall, the manuscripts is well-performed and well-written. The results in a clear manner. The significance of the findings in relation to treating breast cancer is clear. However, there were several issues that need to be addressed to clarify key points of data.

Major concerns:

1. The authors conclude that Brca1 deficiency creates a tumor-permissive microenvironment by increasing MDSCs during tumorigenesis. The data shown in Figure 1 did not support this conclusion. The authors did not compare Brca WT and deficient tumors and the WTMG and MTMG has very minor differences in the myeloid cells. Without this comparison, the only conclusion is that tumorigenesis is increasing MDSCs and it is not dependent on Brca status. Along the same lines: in Figure 1H, the authors should compare the MDSC from WT versus BRCA MT tumor bearing mice to demonstrate if MDSCs from BRCA MT tumor bearing mice are more or equally suppressive.
2. Figure 5H: It would be important to address how overexpressing CXCL12 did not increase S100A9. Does the S100A9/CXCL12 axis not respond if CXCL12 is overexpressed? Similar data shown in Supplemental figure 5H
3. Figure 7M: The authors repeatedly highlight that targeting S100A9 and/or CXCL12 would sensitize tumors to checkpoint inhibitors, but the anti-tumor efficacy of the checkpoint inhibitor alone is missing from this tumor study. Without that checkpoint alone control, it is not possible to conclude that these tumors are being sensitized to checkpoint inhibitors. Additionally, in regards to the data collected on the immune subsets in the tumors (Figure 7O), it would be interesting to see what the immune subsets looks like prior to the tumors being nearly resolved as well as to see the data shown as an average + SD to get a better sense of how consistent the data is. While the myeloid cells were greatly reduced at this later timepoint, it would be interesting to see if they are also altered earlier when the tumor larger (around day 13).

Specific issues:

1. Please include the concentrations used for recombinant S100A9 and tasquinimod for in vitro assays.

2. What dose of PD-1 was used for in vivo experiments?
3. Is the 200mg/mouse dose for tasquinimod and AMD3465 accurate? Please address why such a large dose was used relative to what has been shown in the literature and if there were any toxicities observed with such a high dose.
4. In terms of statistics, One-way ANOVAs should be used in place of unpaired student t-tests when there are more than two groups in a graph.
5. Figure 1D: The title of the graph and the y- axis are not in agreement.
6. Figure 2 real-time PCR: No information on what each dot represents i.e. experiment replicates, biological replicates?
7. Figure 2B & 2E: One-way ANOVA should be used for statistics, not a student t-test.
8. Figure 6A/B graphs: what does each data point represent (experimental replicates, biological replicates)? One-way ANOVA should be performed on biological replicates.
9. Figure 6H & I: Please indicate what cell population the frequency is being measured in? (i.e. total cells, alive cells, CD45 cells)
10. Figure 6J: The staining for dot plot on upper right doesn't look like others. Q1 contains 77.6% of cells instead of the 13-17% that the others show. Please address this.
11. Figure 6K-L: Legend states there were 6-10 mice from each group, yet only 3 points are shown on graph. Please address this discrepancy. Please show the error (SD) on the control group. Further, a one-way ANOVA should be performed for statistical test.
12. Figure 7D: Please clarify what the CD45/CD3e (dark green) population is. The CD8 (bright green) and CD4 (pink) positive CD3 cells are already being accounted for.

Reviewer #2:

Remarks to the Author:

Nice study, well controlled, relatively novel.

Reviewer #3:

Remarks to the Author:

The authors have evaluated the role of S100A and CXCL2 in the context of BRCA1 deficient mammary tumors.

1. In Figure 1E-F, do the authors feel that splenic MDC gene analysis is representative of the tumor-resident MDSC?
2. In Figure H, do the MDSC from tumor-bearing mice behave in a similar manner?
3. In Figure 2F-G, is it possible to determine which cells are expressing S100A9?

4. In Figure 3A, as a control, it might be of interest to see what effect shBRCA2 might have on S100A9/8 expression.

5. In Figure 3E, it is stated that the reason for adding S100A9 cDNA (to increase protein levels) is due to the fact that "accumulated S100a9 protein could induce expansion and activation of MDSCs in both human and mouse (30, 31) and pathologically activate PD1/PD-L1 immune checkpoint." However, it is not clear to the reviewer why this experiment was performed. Please clarify for the reader.

6. In Figure 4, is the expression of S100A9 assumed to be within mammary-derived cells as opposed to infiltrating immune cells?

7. Figure 5: the title of this section is misleading. It should read something like the figure legend, namely "Cxcl12 is upregulated by (or - in response to) elevated S100a9 in Brca1 MT epithelial cells". At this point the authors should also state any evidence linking S100A9 with chemokine expression of any sort.

8. In Figure 5I, it should be mentioned that several receptors beside RAGE have been described for extracellular S100A8 and S1009. These include TLR 4, the scavenger receptor CD36, and EMMPIRIN.

9. Figure 6: Might it be possible to test the culture supernatants against murine myeloid cells from WT and BRCA1 mutant mice?

10. Figure 7. Is there any data on the effect of the two inhibitory drugs (Tas for S100a9, and AMD3465 for CXCR4) on PD-L1 expression by tumors or PD-1 expression by T cells?

11. The evidence for a true S100A9-CCL12 axis is modest. The relationship is more on the order of an interaction. The authors should reconsider how they describe the interplay of these 2 factors.

Reviewer #4:

Remarks to the Author:

This paper "S100A9-CXCL12 Axis in BRCA1 Deficiency Orchestrates the Immunosuppressive Microenvironment to Enable Breast Cancer Resistant to Immune Checkpoint Blockade" deals with a study of immune microenvironment of BRCA1 mutant mammary tissues and tumors derived from both BRCA1 mutant mouse models and human xenograft models to identify intrinsic determinants governing tumor progression and the

responses to immune checkpoint blockade. This paper is based on a proteomic study of mouse mammary gland, tumor and adjacent in different development stages between Brca1-WT and MT by a SWATH-MS approach.

INTRODUCTION

Authors described the contribution and involvement of DNA damage repairs genes in predisposition to breast cancer and many biologic process but with inaccuracies :

Line 74, germline BRCA1 mutation responsible for 20-25% (not 30%) according to Miao et al,2020 and Hall, cancer 2009

Line 75 precision on BRCA1 expression (somatic ?) (in parallel, somatic BRCA1 and BRCA2 mutations represent less than 5% of breast cancers, Bertucci, Nature, 2019)

Due to the fact that you subsequently evoke higher tumor mutational burden, in addition to just evoking the role of BRCA1 in different processes (line 78), it would have been interesting to explain further its role in Homologous recombination defect with genomic instability that results from the preferential use of the non-conservative double stranded DNA breaks repair.

Line 94 : Unless the strong rational to combine PARP inhibitors and ICB especially in HRD tumors, it could been interesting to also underline that in clinical trials, no correlation have been observed between response to immune therapy and TMB. TMB is not correlated to PD-L1 and TILS and benefit to immune therapy was limited in PD-L1 subgroups (Impassion 130 and TONIC trial).

SWATH-MS analyses

The results of this study largely depend on the SWATH-MS analyses but these experiments do not seem to have been carried out correctly. First, in Materials and Methods, no technical parameters are given and the presentation of the results show that the authors have absolutely no knowledge of this approach.

How the samples were processed? How the proteins were extracted and digested? How much peptide was injected? Was a micro or nano LC-MS/MS? What were the SWATH-MS parameters?

Apparently the samples were not injected in duplicate. In these conditions, calculations seem difficult to do. (CV?...)

No DDA files were uploaded and we do not know how the library was calculated?

The results table is even worse. The authors present identified proteins with very bad scores which means that the results of this study are probably all wrong. All calculations were done with Peakview; it would be better to do the calculations with Spectronaut.

From these wrong results, the authors calculate ratio between proteins without pValue.

How can they be sure the results are significant?

The paper is based on the identification and quantification of S100A9 but, I am not sure that this protein was correctly identified and quantified.

Others points

Line 117 precise if tumor and tumor adjacent mammary gland is mutated BRCA1 ?

Line 119 : except MDSCs assessment, no macrophages markers (M1/M2) have been selected to accessed the microenvironment immune response ? (we know the impact in terms of immune response in TNBC, Jezequel et al, 2015 and 2019)

Line 528 : "It has been shown previously that tumors with high TMB, defective HR, and/or increased

529 expression of genes involved in immune checkpoint are more sensitive to ICB." Cf note line 94

This paper do not satisfy the technical conditions for a publication in Nature communications

Responses to Reviewers' comments

Reviewer #1 (Remarks to the Author): with expertise in MDSC

The manuscript entitled “S100A9-CXCL12 axis in BRCA1 deficiency orchestrates the immunosuppressive microenvironment to enable breast cancer resistant to immune checkpoint blockade” identifies a novel connection between BRCA1 loss in breast cancer and elevation of S100A9 and CXCL12 providing a possible mechanism for checkpoint resistance. Upon identifying S100A9, the authors confirm the increased expression in their BRCA MT system and connect elevation of S100A9 to patient data. The authors then examine the BRCA/S100A9/CXCL12 axis in a variety of cell lines with using both CRISPR and OE of the proteins. Furthermore, the authors utilize a variety of in vivo models to provide additional evidence of the BRCA/S100A9/CXCL12 axis. Overall, the manuscripts are well-performed and well-written. The results in a clear manner. The significance of the findings in relation to treating breast cancer is clear. However, there were several issues that need to be addressed to clarify key points of data.

Thanks a lot for your insightful suggestions, which are critical for this study. We have revised our manuscript according to your suggestions, which will be detailed below.

Major concerns:

1. The authors conclude that Brca1 deficiency creates a tumor-permissive microenvironment by increasing MDSCs during tumorigenesis. The data shown in Figure 1 did not support this conclusion. The authors did not compare Brca1 WT and deficient tumors and the WTMG and MTMG has very minor differences in the myeloid cells. Without this comparison, the only conclusion is that tumorigenesis is increasing MDSCs, and it is not dependent on Brca1 status. Along the same lines: in Figure 1H, the authors should compare the MDSC from WT versus BRCA MT tumor bearing mice to demonstrate if MDSCs from BRCA MT tumor bearing mice and more or equally suppressive.

In order to well conclude that Brca1 deficiency creates a tumor-permissive microenvironment by increasing MDSCs during tumorigenesis, we have added Brca1-WT breast tumors (WT-BT) into our revised Fig. 1A-G. These data suggest that Brca1 deficiency could induce the tumor-permissive microenvironment by increasing MDSCs during tumorigenesis.

The old Fig. 1H is now changed Fig. 1L-1O as more comparisons are added (please also see R-Fig. 1A-1D below). We compared the MDSCs from spleen, mammary glands and tumor tissues in Brca1-WT and Brca1-MT tumor bearing mice to investigate whether MDSCs from BRCA1 MT tumor bearing mice had more or equally suppressive T cells. The data revealed that the MDSCs from spleen and mammary gland tissues could suppress T-cell proliferation (R-Fig. 1Aa-c and 1Ca-c) and MDSCs from both spleen and tumor tissues in Brca1-MT mice had much stronger suppressive effects

on T-cell proliferation than those from Brca1-WT mice (R-Fig. 1Ad-e, 1B, 1Cd-e, 1D). We have described information in lines :153-159 on page 5 and included these data in Fig. 1L-O on page 32.

R-Figure 1. Representative CFSE flow-cytometry histograms showing the effect on WT T cell proliferation by MDSCs in vitro from spleen, mammary and tumor tissues of WT and Brca1-MT mice.

(A-B) Representative histogram plots for T cell proliferation from spleen: unstimulated T cells (Aa), stimulated T cells with CD3/CD28 (Ab), T cells were cocultured with MDSCs isolated from spleen of Brca1-MT mice without tumor (Ac), WT tumor bearing mice (Ad), and Brca1-MT tumor bearing mice (Ae) at 1:1 ratio. Quantification (B) for panels in (A).

(C-D) Representative histogram plots for T cell proliferation from mammary and tumor tissues: unstimulated T cells (Ca), stimulated T cells with CD3/CD28 (Cb), T cells were cocultured with MDSCs isolated from mammary gland of Brca1-MT mice without tumor (Cc), from tumor tissue of WT tumor bearing mice (Cd), and from tumor tissue of Brca1-MT tumor bearing mice (Ce) at 1:1 ratio. Quantification (D) for panels in (C) (n=3 mice/group).

2. Figure 5H: It would be important to address how overexpressing CXCL12 did not increase S100A9. Does the S100A9/CXCL12 axis not respond if CXCL12 is overexpressed? Similar data shown in Supplemental figure 5H

We might not indicate it clearly in the previous manuscript. Actually, our experiment indicated that overexpression of CXCL12 increases S100A9 as shown in the original Figure 5H and supplemental figure 5H (red circle showing OE-CXCL12 vs Ctr).

:

Figure 5H: Protein levels of S100a9, Cxcl12, and pStat3 in WT (B477) cells with the expression of OE-S100a9, OE-S100a9/sgCxcl12, and OE-Cxcl12. Supplemental figure 5H: Protein levels of S100a9, Cxcl12, and pStat3 in WT (231) cells with the expression of OE-S100a9, OE-S100a9/sgCxcl12, and OE-Cxcl12.

Regarding your question “Does the S100A9/CXCL12 axis not respond if CXCL12 is overexpressed? Similar data shown in Supplemental figure 5H”, we further investigated the regulation between S100A9 and CXCL12. We collected samples with either overexpression of *S100a9* (R-Fig. 2A, B) or *Cxcl12* (R-Fig. 2C,D) in B477 (New Fig. 5I,J) and MDA-MB231 cells or sgS100a9 in G600 cells (R-Fig. 2E,F); New Fig. 5K,L at different time points and different doses for 48 hours to examine the protein level of CXCL12 or S100A9. We found there was a positive regulation between *S100A9* and *CXCL12* because when we over expressed of either *S100A9* or *CXCL12*, the protein levels of CXCL12 or S100A9 were increased (R-Fig. 2A-D), but the protein levels of S100A9 or CXCL12 were decreased in G600 cells expressing either sg*CXCL12* or sg*S100A9* (R-Fig. 2E-F).

This data demonstrated that there is a positive regulation loop between S100a9 and Cxcl12 which can expand the S100a9 oncogenic signal. We have added description in lines 340-348 on page 11, Fig. 5I, 5J, 5K, and 5L on page 40 and Fig. S5K, and S5L on page 6 in supplementary data.

R-Figure 2. Protein levels of S100A9 and CXCL12 in B477, 231 cells with over expression of either Cxcl12 or S100a9 and G600 cell line with expression of either sgCxcl12 or sgS100a9 at different time point or different amount of DNA at 48 hours after transfection.

(A-B) Protein levels of S100A9 and CXCL12 in B477 (A) and MDA-MB-231 (B) cells expressing *S100A9*. (C-D) Protein levels of S100A9 and CXCL12 in B477 (C) and MDA-MB-231 (D) cells expressing *CXCL12*. (E-F) Protein levels of S100A9 and CXCL12 in G600 either expression of sg*S100a9* (E) or sg*Cxcl12* (F).

3. Figure 7M: The authors repeatedly highlight that targeting S100A9 and/or CXCL12 would sensitize tumors to checkpoint inhibitors, but the anti-tumor efficacy of the checkpoint inhibitor alone is missing from this tumor study. Without that checkpoint alone control, it is not possible to conclude that these tumors are being sensitized to checkpoint inhibitors. Additionally, in regard to the data collected on the immune subsets in the tumors (Figure 7O), it would be interesting to see what the immune subsets looks like prior to the tumors being nearly resolved as well as to see the data shown as an average + SD to get a better sense of how consistent the data is. While the myeloid cells were greatly reduced at this later timepoint, it would be interesting to see if they are also altered earlier when the tumor larger (around day 13).

Thanks a lot for the valuable suggestions. We have added the α PD1 single treatment group in R-Fig. 3, and there is no inhibitory effect in breast tumor from Brca1-MT 545 cells with anti-PD1 treatment only compared to 545-Ctr group (Fig. 7M-N; also see R-Fig. 3A-B). To examine the changes of MDSCs during the tumor progression, we performed CyTOF analysis with antibodies of CD45, CD3e, CD4, CD8a, CD28, CD11B, Ly6G, and Ly6C to tumor tissues at day 13 and day 29 as suggested. We found that the increased CD4, CD8, and CD28 positive T cells and decreased M-MDSC and PNM-MDSC population at day 13 in both α PD1+Tas and α PD1+ADM 3465 groups, but not significant in α PD1 treatment group (Fig. 7Q-R; also see R-Fig. 3C-D), and that the increased T cell and decreased MDSC cell populations were further enhanced at day 29 (R-Fig. 3D). These data were described in lines: 492-494 on page 15 & 505-508 on page 16 and Fig. 7M, 7N, 7Q, and 7R on page 44.

R-Figure 3. Treatment of 545 cells derived breast tumor with PD1 antibody and immune cell status under single and combinatory treatments.

(A-B) Tumor images (A) and volume (B) from fat pad implanted 545 cells in FVB mice (n=7 mice/group).

(C-D) Relative populations of CD4⁺ T cells, CD8⁺ cells, CD28⁺ cells, PNM-MDSC cells, and M-MDSC cells in CD45⁺ cells at day 13 (C) and day 29 (D) without treatment, under treatment with aPD1 only, aPD1+Tas, and aPD1+AMD3465 (n=3 mice/group).

Specific issues:

1. Please include the concentrations used for recombinant S100A9 and tasquinimod for in vitro assays.

The final concentration of recombinant S100A9 is 0.1mg/ml and tasquinimod is 50 μ M. We have included these data in line 1203 of Fig. 5G on page 41 and lines 1230 of Fig. (6I-J.) on page 43.

2. What dose of PD-1 was used for in vivo experiments?

The dose of antibody of PD-1 is 0.2mg/mouse in vivo experiments.

3. Is the 200mg/mouse dose for tasquinimod and AMD3465 accurate? Please address why such a large dose was use relative to what has been shown in the literature and if there were any toxicities observed with such a high dose.

Sorry for the typo, it should be 0.2mg/mouse for tasquiminod and AMD3465. We have revised the information in lines 646-647 on page 20.

4. In terms of statistics, One-way ANOVAs should be used in place of unpaired student t-tests when there are more than two groups in a graph.

Yes, we have re-analyzed the data when there are more than two groups in a graph with One-way ANOVAs..

5. Figure 1D: The title of the graph and the y- axis are not in agreement.

We have made the title of the graph and the y- axis in agreement. We have revised the information at page 32 (Fig.1E, 1G).

6. Figure 2 real-time PCR: No information on what each dot represents i.e. experiment replicates, biological replicates?

Each dot represents biological replicates. We have added the information in line 1138-1139 on page 35 (Fig. 2).

7. Figure 2B & 2E: One-way ANOVA should be used for statistics, not a student t-test.

We have re-analyzed the data with one-way ANOVA analysis.

8. Figure 6A/B graphs: what does each data point represent (experimental replicates, biological replicates)? One-way ANOVA should be performed on biological replicates.

Each data point represents experimental replicates. We repeated this experiment 3 times and performed statistics with all the pictures in these 3 times. We have re-analyzed the data with one-way ANOVA analysis. We have added the description in line 1222 on page 42.

9. Figure 6H & I: Please indicate what cell population the frequency is being measured in? (i.e., total cells, alive cells, CD45 cells)

It is in total cells. We have revised the label in Fig. 6H & I (as Fig. 6M & N in this revision).

10. Figure 6J: The staining for dot plot on upper right doesn't look like others. Q1 contains 77.6% of cells instead of the 13-17% that the others show. Please address this.

We have repeated this experiment and the percentage is about 32%. We have revised figure accordingly (new Fig. 6O).

11. Figure 6K-L: Legend states there were 6-10 mice from each group, yet only 3 points are shown on graph. Please address this discrepancy. Please show the error (SD) on the control group. Further, a one-way ANOVA should be performed for statistical test.

We did this experiment with 6-10 mice in each group and randomly mixed them into 3 samples to do FACS analysis. We have revised the legend states in old Figure 6K-L (now Figure 6Q-R). We have also re-analyzed the data with one-way ANOVA and add the SD in old Figure 6K-L.

12. Figure 7D: Please clarify what the CD45/CD3e (dark green) population is. The CD8 (bright green) and CD4 (pink) positive CD3 cells are already being accounted for.

Sorry, the presentation of old Figure 7D is unclear. We have made a new Fig. 7D with clear presentation of the data.

Reviewer #2 (Remarks to the Author): with expertise in breast cancer and TME

Nice study, well controlled, relatively novel.

Thanks for your approbation and support for this study.

Reviewer #3 (Remarks to the Author): with expertise in breast cancer and MDSC

We would like to thank the reviewer for the insightful suggestions, which are critical for this study. We have revised our manuscript according to your suggestions, which will be detailed below.

The authors have evaluated the role of S100A9 and CXCL12 in the context of BRCA1 deficient mammary tumors.

1. In Figure 1E-F, do the authors feel that splenic MDSC gene analysis is representative of the tumor-resident MDSC?

We feel this is a very good question that needs more data. To provide more evidence, we also added more samples in the analysis, i.e., added mammary gland and mammary tumors in addition to the spleen for MDSC gene analysis from splenic, mammary gland and tumor-resident MDSCs (R-Fig. 4A). PCA analysis revealed that CD11B⁺/GR1⁺ cells could be well separated in 8 groups tissues and samples in each genotype are group together (R-Fig. 4B). However, no matter in the spleen or mammary gland/tumor tissues, the expressions of MDSCs signature genes are enriched with associated with associated with Brca1 mutant status and tumorigenesis by GSEA analysis (R-Fig. 4C-E). We have described this in lines:134-141 on page 5 and included these data in Figs. 1I, 1J, and 1K on page 32 and S1G and in Fig. S1G on page 2 of supplementary Fig.

R-Fig. 4 Enriched MDSC signature genes in CD11b/Gr1 double positive cell population from mammary gland, spleen, tumor tissues between WT and Brca1co/co/MMTV mice by bulk RNA sequences

(A-B) Isolation of MDSCs from spleen and mammary tissues in both WT age matched and *Brca1co/co/MMTV* (MT) mice without tumor, and tumor bearing MT mice (A) and PCA analysis of samples (B) for panel (A) (n=3 mice/group).

(C-E) GSEA analysis of MDSC signature genes by comparing signature genes from the spleen of MT mice without tumor and WT age matched mice (C, left) and MT mice with tumor and MT mice without tumor (C, right), from mammary tissues from MT mice and WT age matched mice without tumor (D, left), MT mice and WT mice both with tumor (D, right), from tumor tissues vs tumor adjacent mammary tissues in MT mice (E, left), and tumor adjacent mammary tissues vs Mammary tissues in MT mice without breast tumor (E, right) (n=3 mice/group).

2. In Figure H, do the MDSC from tumor-bearing mice behave in a similar manner?

To investigate whether MDSC from tumor-bearing mice behave in a similar manner (old Figure 1H is changed to new Figure 1L-O), we isolated MDSCs of the spleen, mammary and tumor tissues from tumor bearing mice in both *Brca1*-WT and *Brca1*-MT mice, and *Brca1*-MT mice without tumor to compare the suppression to T cells (R-Fig. 5A-D). The data showed that MDSCs from both the spleen and mammary tissues of *Brca1*-MT mice behaved similarly in *Brca1*-MT mice without or with tumor in terms of inhibition for T cell proliferation but much stronger inhibition of MDSCs from *Brca1*-MT tumor bearing mice to T cells was observed because the T-cell proliferations were reduced from 6.09% to 0.67% in spleen (R-Fig. 5Ac-e) and from 5.25% to 0.648% in breast tumor tissues (R-Fig. 5Cc-e). We have included description lines:153-159 on page 5 and the data into Figure 1L-O on page 32.

R-Fig. 5. Representative CFSE flow-cytometry histograms showing the effect on WT T cell proliferation by MDSCs in vitro from spleen, mammary and tumor tissues of WT and Brca1-MT mice.

(A-B) Representative histogram plots for T cell proliferation from spleen: unstimulated T cells (Aa), stimulated T cells with CD3/CD28 (Ab), T cells were cocultured with MDSCs isolated from spleen of *Brca1*-MT mice without

tumor (Ac), WT tumor bearing mice (Ad), and Brca1-MT tumor bearing mice (Ae) at 1:1 ratio. Quantification (B) for panels in (A).

(C-D) Representative histogram plots for T cell proliferation from mammary and tumor tissues: unstimulated T cells (Ca), stimulated T cells with CD3/CD28 (Cb), T cells were cocultured with MDSCs isolated from mammary gland of Brca1-MT mice without tumor (Cc), from tumor tissue of WT tumor bearing mice (Cd), and from tumor tissue of Brca1-MT tumor bearing mice (Ce) at 1:1 ratio. Quantification (D) for panels in (C) (n=3 mice/group).

3. In Figure 2F-G, is it possible to determine which cells are expressing S100A9?

To determine which cells are expressing S100A9, we co-stained S100A9 with either K18, a marker for luminal epithelial cells, or CD206, a marker for M2 like macrophage (one type of immune cells) on the tissue sections from the same samples in old Fig.2F-G (new Fig.2H-2I). We detected colocalization of S100A9 with either CK18 or CD206 protein on BRCA1-MT (R-Fig. 6A-B), but not in BRCA1-WT breast cancer samples (R-Fig. 6C-D), showing that mammary epithelial cells had higher levels of S100A9 and more S100A9 positive cells myeloid cells in BRCA1 deficient cancer mammary tissues. We have described above information in lines: 204-207 on page 7 and included these data Fig. S2G-S2H) on page 3 in Supplementary data.

R-Fig. 6. The representative images of mammary tumor tissues from PDX mouse model were co-stained with antibodies of S100A9 and K18 or S100A9 and CD206 by IF.

(A-B) Representative images of breast cancer tissues co-stained with S100A9 and CK18 (A) or S100A9 and CD206 (B) by IF. The representative cells were indicated by arrowhead. The breast cancer tissues were from PDX mouse model from **Jackson laboratory** which were derived from breast cancer patients with BRCA1 mutation.

(C-D) Representative images of breast cancer tissues co-stained with S100A9 and CK18 (C) or S100A9 and CD206 (D) by IF. The representative cells were indicated by arrowhead. The breast cancer tissues were from PDX mouse model from **Jackson laboratory** which were derived from breast cancer patients without BRCA1 mutation.

4. In Figure 3A, as a control, it might be of interest to see what effect shBRCA2 might have on S100A9/8 expression.

As suggested, we expressed shBRCA2 in B477 and MDA-MB231 cell lines. As shown from following, there was no significant effect of shBRCA2 at mRNA expressions (R-Fig.7A-B) and protein levels of S100A9/8 (R-Fig. 7C-D). This description was added in lines:220-221 on page 7 and the data was included in Fig. S3A-D on page 4 in supplementary data.

R-Fig. 7. The mRNA and protein levels of S100A9/8 between Ctr and shBRCA2 in B477 and 231 cell lines
(A-B) Relative expressions of S100a8, S100a9, and Brca2 at mRNA levels in B477 (A) and MDA-MB-231 cells (B) without or with expressions two different shBrca2 vectors.
(C-D) Protein levels of S100A8, S100A9, and BRCA2 from B477 (C) and MDA-MB-231 cells (D) without or with expressions two different shBRCA2 vectors by Western blots.

5. In Figure 3E, it is stated that the reason for adding S100A9 cDNA (to increase protein levels) is due to the fact that “accumulated S100a9 protein could induce expansion and activation of MDSCs in both human and mouse (30, 31) and pathologically activate PD1/PD-L1 immune checkpoint.” However, it is not clear to the reviewer why this experiment was performed. Please clarify for the reader.

We would like to know if S100A9 could positively regulate itself and form the positive feedback loop. We have revised the statement as the following: “Because accumulated S100a9 protein could induce expansion and activation of MDSCs and activated MDSCs can secrete S100A9 (33, 34), we next want to explore if S100a9 could regulates itself to form a positive feedback loop.” in lines: 234-236 on page 7-8.

6. In Figure 4, is the expression of S100A9 assumed to be within mammary-derived cells as opposed to infiltrating immune cells?

S100A9 is expressed at lower level in the mammary epithelial cells and high level in the infiltrating immune cells. In Fig. 4A, we first sorted those populations by FACS with CD24 and CD29 antibodies, and then used qRT-PCR to determine the expression of S100A9 in mammary-derived cells (Fig. 4 left) and stromal cells (Fig. 4 right). Higher S100A9 protein levels were detected in Brca1-MT mammary epithelial cells at different development stages (Fig. 4C). The infiltrating immune cells were also enriched in Brca1co/co/MMTV mutant mice based on following observations: 1) gradually increased S100A9 positive cells in blood of Brca1co/co/MMTV mice from 4-months to 6-months compared to WT mice by FACS analysis in (Fig. 4D). 2) The invasion of MDSCs onto mammary gland tissues was often observed in Brca1co/co/MMTV mice not in WT mice (Fig. 4E right), 3) S100A9 positive cells could be co-localized with M2 like macrophage marker CD206 in Brca1-MT mammary tissues (Figs. S2G, 4E left), but not in WT mice (Fig. S2H), showing that loss function of Brca1 enhanced S100a9 expression in mammary epithelial cells and enriched s100A9 MDSC cells in mammary tissues.

7. Figure 5: the title of this section is misleading. It should read something like the figure legend, namely “Cxcl12 is upregulated by (or - in response to) elevated S100a9 in Brca1 MT epithelial cells”. At this point the authors should also state any evidence linking S100A9 with chemokine expression of any sort.

Thank you very much for the suggestion. After careful consideration, we decided to change the title of this section to “Positive regulation loop between S100a9 and Cxcl12 amplifies oncogenic signals in Brca1-MT epithelial cells”.

We have included more experiment to study the regulation between S100a9 and Cxcl12 by overexpression of either *S100a9* (R-Fig. 8A,B) or *Cxcl12* (R-Fig. 8C,D) in B477 cells and MDA-MB231 cells or sgS100a9 in G600 cells (R-Fig. 8E,F) at different time points and different doses for 48 hours to examine the protein level of CXCL12 or S100A9. We found there was a positive regulation between *S100A9* and *CXCL12* because when we over expressed of either *S100A9* or *CXCL12*, the protein levels of CXCL12 or S100A9 are increased (R-Fig. 8A-D), but the protein levels of S100A9 or CXCL12 are decreased in G600 cells expressing either sg*CXCL12* or sg*S100A9* (R-Fig. 8E-F). We have added description in lines 340-348 on page 11, Fig. 5I, 5J, 5K, and 5L on page 40 and Fig. S5K, and S5L on page 6 in supplementary data.

R-Fig. 8 Protein levels of S100A9 and CXCL12 in B477, 231 cells with over expression of either Cxcl12 or S100a9 and G600 cell line with expression of either sgCxcl12 or sgS100a9 at different time point or different amount of DNA at 48 hours after transfection.

(A-B) Protein levels of S100A9 and CXCL12 in B477 (A) and MDA-MB-231 (B) cells expressing *S100A9*. (C-D) Protein levels of S100A9 and CXCL12 in B477 (C) and MDA-MB-231 (D) cells expressing *CXCL12*. (E-F) Protein levels of S100A9 and CXCL12 in G600 either expression of sg*S100a9* (E) or sg*Cxcl12* (F).

For investigating the potential relationship between S100A9 and chemokines, we examined expressions of 10 cytokines in B477 and G600 cells (R-Fig. 9A). Our data detected only *S100a9* and *Cxcl12* showed high expressions in Brca1-MT (G600) mammary epithelial cells, but not in Brca1-WT B477 cells, suggesting that the

increased S100a9 and Cxcl12 is associated with Brca1 deficiency. The description has been added into lines: 309-312 on page 10 and the data has been added into Fig. S5A on page 6 in supplementary data.

R-Fig. 9 Expression of cytokines in B477 and G600 mammary epithelial cell lines

(A) Relative expression of any sort of chemokine in B477 and G600 as determined by qPCR.

8. In Figure 5I, it should be mentioned that several receptors beside RAGE have been described for extracellular S100A8 and S1009. These include TLR 4, the scavenger receptor CD36, and EMMPIRIN.

Thanks a lot for the suggestion. We have included a statement that “It has been shown that several receptors beside RAGE have been described for extracellular S100A8 and S1009, including TLR 4, the scavenger receptor CD36, and EMMPIRIN (41). Whether or not these receptors play a role in S100a9-Cxcl12-pStat3 signaling is currently unclear but may be studied in future study”. In Lines: 333-336. on page10-11.

9. Figure 6: Might it be possible to test the culture supernatants against murine myeloid cells from WT and BRCA1 mutant mice?

As suggested, we isolated MDSCs from the spleens of WT mice and Brca1-MT mice with antibodies of CD11b/Gr1 to examine their migration abilities. As shown in the R-Fig. 10A, the number of migrating MDSCs was increased in the conditional mediums from both B477 cells with OE-S100a9 (OE-S100a9-B477) and Brca1-MT G600 cells (Ctr-G600) compared to the control cells (Ctr-B477) (R-Fig. 10Aa-c, 10Be-g, 10C). The number of migrating MDSCs was dramatically decreased if the conditional medium was from G600 cells with expression of sgS100a9 (R-Fig. 10Ad, 10Bh). These data indicated that high level of S100A9 protein in culture medium could affect the MDSC cell migration from both WT and Brca1-MT mice and the effects were stronger on MDSCs from Brca1-MT mice (R-Fig. 10Ab vs 10Bf, 10Ac vs 10Bg, and 10C). We have included this data in lines 361-372 on page 11-12 and experiment data in Figure 6A-C on page 42.

R-Figure 10. Migration assay of MDSCs from WT and Brca1-MT mice spleens under effects of conditional medium from cells with different genotypes.

(A-B) Number of migrating cells on the membranes with different conditional medium used to culture MDSCs from both WT (A) and Brca1-MT mice (B). The conditional medium from WT-B477 cell only (a,e), WT-B477 cells with OE-S100a9 (b,f), G600 cells only (c,g), and G600 cells with the expression of sgS100a9 (d, h).

(C) Quantification for panels in (A-B).

10. Figure 7. Is there any data on the effect of the two inhibitory drugs (Tas for S100a9, and AMD3465 for CXCR4) on PD-L1 expression by tumors or PD-1 expression by T cells?

Currently, we did not find any data to show this. To test the effects of the two drugs (Tas and AMD3465) for PDL1 expression in tumor cells, we co-stained K18 and PDL1 on breast tumor tissues from *Brca1^{Co/Co};MMTV-Cre* mice: Ctr (untreated), Tas or AMD 3465 treatment group and found PDL1 level in tumor tissues was upregulated after the single treatment with either Tas or AMD3465 (R-Fig. 11A-B). Meanwhile, PD1 expression in T cells was also tested by co-staining CD3 and PD1 on spleen tissues of *Brca1^{Co/Co};MMTV-Cre* mice. The results showed that PD1 level in T cells was also upregulated after treated with either Tas or AMD3465 only compared to the Ctr group (untreated) (R-Fig. 11C-D). These results indicate that single treatment with either Tas or AMD3465 may change the status of tumor cells and benefit combination treatment with either Tas or AMD3465 together with PD1 antibody. We have included the description in lines 494-497 on page 15 and data in Fig. S7M-P on page 8 in supplementary data.

R-Figure 11. CK18 and PDL1 antibody staining on mammary tumor tissues and CD3 and PD1 antibody staining on T cells of *Bra1co/co/MMTV* mice treated with Tas and AMD3465.

(A-B) Representative images of breast tumor tissues of *Bra1co/co/MMTV* mice from untreated group (Ctr), Tas inhibitor treatment group, and AMD3465 inhibitor treatment group co-stained with CK18 and PDL1 antibodies (n=3 mice/group). Single and double positive cells were indicated by arrowhead. Quantification for CK18 and PDL1 double positive cells (B) in panels (A).

(C-D) Representative images of CD3 and PD1 cells from the same cohort of mice co-stained with CD3 and PD1 antibodies and single and double positive cells were indicated by arrowhead. Quantification for CD3 and PD1 double positive cells (D) in panels (C).

11. The evidence for a true S100A9-CXCL12 axis is modest. The relationship is more on the order of an interaction. The authors should reconsider how they describe the interplay of these 2 factors.

To strengthen the axis of S100A9-CXCL12, we over expressed S100a9 and/or Cxcl12 in WT-B477 and MDA-MB-231 cells and disrupted either S100a9 and/or Cxcl12 in G600 cells and collected samples at increasing doses and also at different time point within 48 hours after the transfection to monitor the change of these two proteins. We have also transfected *sgS100a9* and *sgCxcl12* at different concentrations to examine the changes of protein levels of CXCL12 and S100A9. The result of these experiments were shown in R-Fig 8.

Reviewer #4 (Remarks to the Author): with expertise in proteomics

This paper “S100A9-CXCL12 Axis in BRCA1 Deficiency Orchestrates the Immunosuppressive Microenvironment to Enable Breast Cancer Resistant to Immune Checkpoint Blockade” deals with a study of immune microenvironment of BRCA1 mutant mammary tissues and tumors derived from both BRCA1 mutant mouse models and human xenograft models to identify intrinsic deterrents governing tumor progression and the responses to immune checkpoint blockade. This paper is based on a proteomic study of mouse mammary gland, tumor and adjacent in different development stages between *Bra1-WT* and *MT* by a SWATH-MS approach.

Thanks a lot for your insightful and professional suggestions, which are critical for this study. We calculated previous SWATH data with Spectronaut as suggested. We have also collected new samples for Data-independent acquisition mass spectra (DIA-MS) analysis with duplication injection for each sample. We have revised the manuscript accordingly, which will be detailed below.

INTRODUCTION

Authors described the contribution and involvement of DNA damage repairs genes in predisposition to breast cancer and many biologic processes but with inaccuracies:

Line 74, germline BRCA1 mutation responsible for 20-25% (not 30%) according to Miao et al,2020 and Hall, cancer 2009

We have revised 30% to 20-25% in line 75 on page 3.

Line 75 precision on BRCA1 expression (somatic?) (in parallel, somatic BRCA1 and BRCA2 mutations represent less than 5% of breast cancers, Bertucci, Nature, 2019) Due to the fact that you subsequently evoke higher tumor mutational burden, in addition to just evoking the role of BRCA1 in different processes (line 78), it would have been interesting to explain further its role in Homologous recombination defect with genomic instability that results from the preferential use of the non-conservative double stranded DNA breaks repair.

Thank you very much for raising this point. We have highlighted a role of BRCA1 in HR mediated DSB repair in lines 81, and 88-92, and also cited Bertucci, Nature, 2019 (reference 16).

Line 94: Unless the strong rational to combine PARP inhibitors and ICB especially in HRD tumors, it could be interesting to also underline that in clinical trials, no correlation have been observed between response to immune therapy and TMB. TMB is not correlated to PD-L1 and TILS and benefit to immune therapy was limited in PD-L1 subgroups (Impassion 130 and TONIC trial).

It is a good point. Yes, TMB alone does not always predict ICB response. Recent studies have suggested that alterations in the DNA damage response (DDR) pathway, like HR alteration (often due to genetic alterations affecting the core genes BRCA1 or BRCA2) influences response to ICB (Ma et al. 2018. Nat. Common. 9, 3292). We have deleted TMB in line 95, and cited Ma et al. 2018. Nat. Common. 9, 3292 as reference 26.

SWATH-MS analyses

The results of this study largely depend on the SWATH-MS analyses, but these experiments do not seem to have been carried out correctly. First, in Materials and

Methods, no technical parameters are given, and the presentation of the results show that the authors have absolutely no knowledge of this approach.

How the samples were processed? How were the proteins extracted and digested (presented)? How much peptide was injected? Was a micro or nano LC-MS/MS? What were the SWATH-MS parameters?

The previous SWATH data was analyzed by using peakview v2.1 (AB SCIEX). We have now re-analyzed with Spectronaut as suggested. Minimum number of 3 fragment ions and 3 precursor ions with >90% confidence was used for SWATH-MS/MS data processing by National Center for Protein Sciences (Beijing). This analysis identified a total of 4542 proteins previously. We later identified a total of 5243 proteins using Spectronaut approach, of which 322 genes were differentially expressed during the tumorigenesis. S100A8 and S100A9, ARL8B, and CTNBL1 were sorted out as top 4 candidates after overlapping with top 45 human candidate genes which are negatively correlated with BRCA1 expression (R-Fig. 12).

R-Fig. 12 Identify candidate genes from SWATH-MS analysis.

However, since above experiment only used single injection to obtain the data, we then collected new samples and using Data-independent acquisition (DIA) with duplication injection for each sample (n=3 mice/group) by BiotechPack SCIENTIFIC company in this revision. This new analysis also identified 4 top candidate genes, S100A8 and S100A9, Pglyrp1 and Colgalt1 (new Fig S2B, and Fig. 2G). We have mentioned in lines 184-185 that “While all the 4 top candidates might be potentially important, we first investigated S100a8 and S100a9, as our CyTOF data implicating these them in TME. We have described DIA-MS in detail in lines: 696-764 from page 21-23 in material methods).

Mass Spectrum of DIA

1) Sample preparation

Fifteen samples were collected from WT mammary gland (WTMG) -virgin 8-10 month, MT mammary gland (MTMG) -virgin 8–10-month, WT breast tumors (WTBT), Brca1-MT tumors (Brca1-MTBT) and adjacent (BRCA1-MTBT adj.-MG) (n=3 mice/group). The tissues were digested into single cells (protocol in material methods-Cytiof part). The single cell samples from above groups (15 samples) were processed followed procedure in material methods on lin lines: 704-717 from page 22.

2) Spectral Library Generation

The details of Data Dependent Acquisition (DDA) adoption see material methods on lines: 718-743 from page 22-23. Twelve fractions from above mix digested peptides loaded onto Nano LC connected to an electrospray ionization mass spectrometer/ESI Mass Spectrometer (Orbitrap Eclipse Mass Spectrometer-Thermo Fisher Scientific-USA) to adopt in a DDA model. Files from above 12 fractions were imported into Proteome Discoverer (Version 2.4 Thermo Scientific) to search and identify proteins to build the spectral library for quantitative analysis. Search parameters can be found in material methods inlines: 752-757 on page 23.

3) Data Independent Acquisition (DIA) and DIA Analysis

The peptides of 15 samples (5 µg each) were dissolved in the sample solution (0.1%FA, 2% ACN) and adopted in a DIA model. Each sample was injected in duplicate technical replicate (detail procedure in material methods) on page 23. Spectronaut software was used to construct the data and then the collected DIA raw data were compared with the quantitative database to extract the quantitative chromatographic peak area of unique peptide of each protein which was used as the relative abundance characterization of protein. Total 8766 proteins were identified, and the quantitative peptide false discovery rate (FDR) is < 1%. The detail procedures can be found in material methods lines: 744-764 on page 23.

Apparently, the samples were not injected in duplicate. In these conditions, calculations seem difficult to do. (CV?).

In order to calculate CV, we used new samples with duplication injection to do DIA analysis. As shown in follow R-Figure 13A, the CV% of all 15 samples are less than 20% and the median CV% are between 0.8-1.21(R-Fig. 13A) At the same time, Pearson correlation analysis shows that Pearson coefficient between duplicate of one sample is more than 0.90 (R-Fig. 13B). These data confirm that the machine is stable. We have included this data in lines: 168-173 on page 6; Figs. S2B on page 3 in supplementary, and 2C and page 34.

R-Figure 13. CV% and Pearson correlation analysis of duplicate of each sample of 15 samples to test the stability of machine.

(A) CV% of duplicate of each sample of 15 samples.

(B) Pearson correlation analysis of duplicate of each sample of 15 samples.

No DDA files were uploaded, and we do not know how the library was calculated?

We have uploaded the SWATH-DDA files and DIA-DDA files described in Data availability part. (The description will be included in Data availability in lines: 915-917 on page 28).

Below are the proteomics files Submission details:

(1): SWATH-DDA file submission details:

Website: <http://www.ebi.ac.uk/pride>

Project Name: Mouse mammary gland, tumor and adjacent in different development stages between Brca1-WT and MT by SWATH-MS-2(library files-DDA files)

Project accession: PXD030204

Reviewer account details:

Username: reviewer_pxd030204@ebi.ac.uk
Password: T13pruDI

(2) DIA-MS file submission details:

DIA-MS (DIA raw data) : Project Name: The protein profile of mammary gland and tumors between Brca1-WT and MT mice by DIA-MS (DIA-data)

Project accession: PXD030328

Reviewer account details:

Username: reviewer_pxd030328@ebi.ac.uk

Password: N3QaMMHQ

Website: <http://www.ebi.ac.uk/pride>

DIA-MS (DDA raw data) : Project Name: The protein profile of mammary gland and tumors between Brca1-WT and MT mice by DIA-MS (DDA data)

Project accession: PXD030355

Reviewer account details:

Username: reviewer_pxd030355@ebi.ac.uk

Password: K1mGk3qk

Website: <http://www.ebi.ac.uk/pride>

The results table is even worse. The authors present identified proteins with very bad scores which means that the results of this study are probably all wrong. All calculations were done with Peakview; it would be better to do the calculations with Spectronaut.

We have re-analyzed the SWATH data with Spectronaut as suggested and identify the protein with good score (FDR<1%). 5413 proteins were identified with Spectronaut (FDR<1%). Since we did the experiments with new samples, we did not include previous SWATH data into this manuscript.

From these wrong results, the authors calculate ratio between proteins without pValue. How can they be sure the results are significant? The paper is based on the identification and quantification of S100A9 but, I am not sure that this protein was correctly identified and quantified.

As mentioned earlier, we now only use new data generated by DIA with duplication injection to calculate CV% in these five group samples. We selected candidate genes by Pearson r each > 0.5 , adjust p value < 0.05 from website (<https://www.metaboanalyst.ca/faces/home.xhtml>). Because it is a critical point, we would like to show details below although we have described them in the corresponding text and supplemental materials.

As shown in R-Fig. 14A, 632 genes ($r > 0.5$, adjust $p < 0.05$) with differential expressions in MTMG, Adj. tumor tissues, MTBT compared to WTMG and 725 genes

with (FC >2, P < 0.05) significantly upregulated in Brca1-MTBT compared with WTBT were sorted out in DIA analysis. In DIA analysis, S100A9, S100A8, PGLYRP1 and COLGALT1 are four common genes by comparison of 632 and 725 mouse candidate gene lists with human top 45 genes which are negatively correlated with BRCA1 expression from human proteomics database (Figure R-14A). In previous SWATH analysis, S100A9, S100A8, ARL8B and CTNNBL1 were four common genes by overlapping 322 mouse candidates with top 45 human BRCA1 negatively correlated genes (Fig. R-14A). Since S100A9 and S100A8 were sorted out as top candidates from two independent analysis, we feel we had identified right candidate for this study. To further qualitative and quantitative S100a9 peptides in DIA analysis, the enrichment of S100a9 peptides were observed (14-40) during the course of tumorigenesis (R-Fig. 14B) and the increased pattern of more identified S100a9 peptides is significant in MTMG, Adj. tumor tissues, and MTBT compared to WTMG ($r=0.8343$, adjust p value=0.024) (R-Fig.14C). The protein level of S100a9 gradually increase following WTMG-MTMG-Adj.-MTBT pattern ($r=0.8343$, adjust p value=0.024) (R-Fig.14D). This information has been described in lines: 173-189 on page 6 and the data was included into Fig. 2D, 2G on page 34 and Fig. S2B, S2C on page 3 in supplementary data.

A

B

R-Figure 14. S100A9 is picked out as the top candidate gene by DIA and SWATH analysis.

(A) The samples and the strategy of DIA and SWATH analysis. SWATH is from last time, we re-analyzed SWATH data with Spectronaut. DIA is from this time. Result of B is from DIA analysis.

(B-C) Identified numbers of S100A9 peptides in each group (B), and statistics analysis (C) from the same cohort of samples with duplicate injection for each sample in Fig. S1A.

(D) S100A9 gradually increase follow WTMG, MTMG, Adj.tumor tissues, and MTBT pattern, and the increased protein level of S100A9 is significant ($r=0.8343$, adjust p value=0.024).

Others points

Line 117 precise if tumor and tumor adjacent mammary gland is mutated BRCA1?

Based on our previous data, *Brca1* can be disrupted in up to 90% of mammary epithelial cells in the *Brca1^{Co/Co};MMTV-Cre* mouse model (Xu et al. 1999, Nature Genetics, 22, 37-43). We have cited this paper as reference 8 in this study. To confirm the genotype of our samples, including WTMG, *Brca1* MTMG, *Brca1* MTBT-adjacent, WTBT and *Brca1* MTBT, the primer-F/R was used to do PCR and from the following figure-R-Figure 15, we can see *Brca1* have been deleted in *Brca1* MTMG, *Brca1* MTBT-adjacent and *Brca1* MTBT.

R-Figure 15. The schematic diagram of *Brca1*-MT mouse model achieved by MMTV-Cre mediated exon 11 deletion and genotype identification by the primer-F/R in WTMG, *Brca1* MTMG, *Brca1* MTBT-adjacent,

WTBT and Brca1 MTBT groups (WT band- no band between 500-700 bps, >3.4kb band can't been PCR amplified by F/R, and MT band-band between 500-700 bps).

Line 119: except MDSCs assessment, no macrophages markers (M1/M2) have been selected to access the microenvironment immune response? (we know the impact in terms of immune response in TNBC, Jezequel et al, 2015 and 2019)

To provide information about M1/M2 macrophages in WTMG, Brca1-MTMG, Brca1-MTBT-Adj., WTBT and Brca1-MTBT, we used F4/80-CD86 as the M1 macrophage marker and F4/80-CD206 as the M2 macrophage marker and did co-staining by IF. The data revealed that M1 like macrophages with double positive markers of F4/80 and CD86 only showed little increase infiltration in Brca1-MT mammary glands (MTMG, $p > 0.05$) than that in WT, but decrease in Brca1-MT tumor adjacent, and tumor mammary tissues (R-Fig. 16A,C). In contrast, M2 like microphages with double positive markers of F4/80 and CD206 were increased much more in Brca1-MT mammary gland, tumor adjacent and tumor tissues than that in WT mammary gland and tumor tissues, respectively (R-Fig. 16B,D). We have included this data in lines: 141-148 on page 5 and in (Fig. S1H-K) on page 2 in supplementary data. .

R-Figure 16. The representative images of IF staining with M1 and M2 like macrophage markers on mammary tumor tissues at different tumor development stages

(A) Representative images of breast tumor tissues co-stained with antibodies of F4/80 and CD86 which are markers for M1 like macrophage in WTMG, MTMG, Brca1-MT tumor adjacent MG, Brca1-WT breast tumor tissues, and Brca1-MT breast tumor tissues.

(B) Representative images of breast tumor tissues co-stained with antibodies of F4/80 and CD206 which are markers for M2 like macrophage in WTMG, MTMG, Brca1-MT tumor adjacent MG, Brca1-WT breast tumor tissues, and Brca1-MT breast tumor tissues.

(C-D) Quantification of M1 like macrophages (C) for panels in (A) and quantification of M2 like macrophages (D) for panels in (B).

Line 528 : "It has been shown previously that tumors with high TMB, defective HR, and/or increased 529 expression of genes involved in immune checkpoint are more sensitive to ICB." Cf note line 94.

As indicated earlier for answering related question, TMB alone does not always predict ICB response, we would like to delete TMB in the sentence. We have changed in line 578-579, on page 18.

Reviewers' Comments:

Reviewer #1:

Remarks to the Author:

It seems authors made a concerted effort to address concerns raised by the reviewers. They addressed questions thoroughly and satisfactorily. I don't have additional concerns.

Reviewer #3:

Remarks to the Author:

In Figure 1N, it appears that the level of inhibition by MDSC from mutant breast cancers is very similar to that of MDSC from wild-type breast cancers. Therefore the declaration that "MDSCs from both spleen and tumor tissues in Brca1-MT mice had much stronger suppressive effects on T-cell proliferation than those from Brca1-WT mice" should be softened.

With respect to Figure 5H, the degree of increase of S100A9 with CXCL12 in B477 cells is modest. The claims related to this immunoblot should be softened in text of the Results section. Also, in Suppl. Fig 5H why is S100A9 not expressed in the OE-S100A9/sgCxcl12 line (MDA-MB-231)?

Again, the evidence for a true S100A9-CXCL12 axis is based primarily on in vitro data on two cell lines with a minimal explanation of the mechanism involved. Therefore, while the relationship is important, the term "axis" might best be replaced with a word such as "interaction".

Reviewer #4:

Remarks to the Author:

The authors took into account my comments.
I support the publication of this article.

Responses to Reviewers' comments

1) In Figure 1N, it appears that the level of inhibition by MDSC from mutant breast cancers is very similar to that of MDSC from wild-type breast cancers. Therefore, the declaration that “MDSCs from both spleen and tumor tissues in Brca1-MT mice had much stronger suppressive effects on T-cell proliferation than those from Brca1-WT mice” should be softened.

Response to the question 1: We have softened the claim by replace it with the following sentence: “MDSCs from both spleen and tumor tissues in Brca1-MT mice had more suppressive effects on T-cell proliferation than those from Brca1-WT mice” in lines 135-138, on page 5.

2) With respect to Figure 5H, the degree of increase of S100A9 with CXCL12 in B477 cells is modest. The claims related to this immunoblot should be softened in text of the Results section. Also, in Suppl. Fig 5H why is S100A9 not expressed in the OE-S100A9/sgCxcl12 line (MDA-MB-231)?

Response to question 2: in Figure 5h, the effect is moderate. We have softened claim for Figure 5h, and the effect of increased protein levels of S100A9 induced by overexpression of *Cxcl12* compared with control were observed in both B477 and 231 cells (Fig. 5j; Supplementary Fig. 5i) in lines 319-322, on page 10.

For Supplementary Figure 5H: This is because *cxcl12* and *s100a9* form the positive feedback loop, so knockout of *cxcl12* affects *s100a9* expression in MDA-MB-231 cells.

3) Again, the evidence for a true S100A9-CXCL12 axis is based primarily on in vitro data on two cell lines with a minimal explanation of the mechanism involved. Therefore, while the relationship is important, the term “axis” might best be replaced with a word such as “interaction”.

Response to question 3: Answer: We have changed “axis” to “interaction” in line 332 on page 10.